# Species-Specific Flash Patterns Track the Nocturnal Behavior of Sympatric Taiwanese Fireflies

**DOI:** 10.3390/biology11010058

**Published:** 2022-01-01

**Authors:** King-Siang Goh, Chia-Ming Lee, Tzi-Yuan Wang

**Affiliations:** 1Genomics Research Center, Academia Sinica, Nankang, Taipei 115, Taiwan or gohks@gate.sinica.edu.tw; 2Research Center for Applied Sciences, Academia Sinica, Nankang, Taipei 115, Taiwan; charlee@gmail.com or; 3Biodiversity Research Center, Academia Sinica, Nankang, Taipei 115, Taiwan

**Keywords:** firefly, sympatric population, flash pattern, flash interval, flash duration, species identification, nightly active-period, *Abscondita cerata*, *Luciola kagiana*, *Luciola curtithorax*

## Abstract

**Simple Summary:**

Nocturnal fireflies are facing various challenges to their survival (e.g., artificial light pollution and extreme climate change). However, there are still no effective diagnostic tools to distinguish firefly species and monitor how their population size and nocturnal behavior change in the wild. Here, we demonstrated that flash interval (FI), a timing character of firefly flash signaling, can be a reliable species-specific luminescent marker for identifying three species (*Abscondita cerata*, *Luciola kagiana*, and *Luciola curtithorax*) in a wild sympatric population and evaluating their nightly behaviors in video-imagery analysis, as well as a traditional field specimen collection. Our studies provide key information and valuable references for future studies of (1) the behaviors of sympatric fireflies in the wild and (2) the development of automatic and rapid digital diagnostic tools for identifying firefly species and quantifying population size, especially for firefly conservation.

**Abstract:**

It is highly challenging to evaluate the species’ content and behavior changes in wild fireflies, especially for a sympatric population. Here, the flash interval (FI) and flash duration (FD) of flying males from three sympatric species (*Abscondita cerata*, *Luciola kagiana*, and *Luciola curtithorax*) were investigated for their potentials in assessing species composition and nocturnal behaviors during the *A. cerata* mating season. Both FI and FD were quantified from the continuous flashes of adult fireflies (lasting 5–30 s) via spatiotemporal analyses of video recorded along the Genliao hiking trail in Taipei, Taiwan. Compared to FD patterns and flash colors, FI patterns exhibited the highest species specificity, making them a suitable reference for differentiating firefly species. Through the case study of a massive occurrence of *A. cerata* (21 April 2018), the species contents (~85% of the flying population) and active periods of a sympatric population comprising *A. cerata* and *L. kagiana* were successfully evaluated by FI pattern matching, as well as field specimen collections. Our study suggests that FI patterns may be a reliable species-specific luminous marker for monitoring the behavioral changes in a sympatric firefly population in the field, and has implication values for firefly conservation.

## 1. Introduction

Among the various glowing organisms (e.g., bacteria, jellyfish, fungi), fireflies (Coleoptera: Lampyridae) are one of the best-known bioluminescent creatures due to their amazing flash communications during courtship [1]. For adult fireflies, the luminous signaling is more than a courtship signal, but also can act as a warning or luring signal [2,3]. Adult fireflies emit light via a specialized abdominal light organ where luciferase–luciferin-dependent luminescent reaction occurs [4]. The adult light organ can generate the light into discrete rapid flashes, which are controlled by nitric oxide (NO) released from the nerve cells via oxygen-gating to photocytes [4,5]. Over the past century, fireflies and their luminosity have inspired numerous scientific studies and industrial applications, such as improving the light intensity of light-emitting diodes (LEDs) and using luciferases as a diagnostic genetic reporter [6,7]. Recently, the genomes of several fireflies were completely sequenced, lighting a new path towards understanding the molecular mechanism behind firefly bioluminescence [8]. Although fireflies’ luminous courtships have been intensively studied for decades [3,9], their nightly behaviors remain a mystery. For instance, do they glow or flash throughout the entire night? Do they mate during a specific time of night? How do distinct firefly species cohabit without interfering with each other’s luminous communications?

To date, more than 2000 firefly species have been described globally, from tropical to temperate zones [10]. All fireflies are luminescent during their larval stage (1–2 years), and the luminous courtship behavior only occurs during the short adult stage (2–4 weeks) [11,12]. To prevent cross-species mating, each firefly species has its own species-specific luminescence features (e.g., flash durations, flash frequencies, and flash colors) [3]. All known luminescent signaling of adult fireflies is roughly classified into the discrete flashing-form and continuous glowing-form [13,14]. Most firefly species use the flashing-form for signaling. Synchronous flashing is the best-known group behavior, but it only occurs in some species (members of *Pteroptix*, *Photinus*, and *Luciola*) while they are congregated [15]. It is believed that this behavior plays a role in eliminating visual clutter to increase successful mating [16]. Compared with synchronous flashing, studies of non-synchronous flashing behavior are still rare.

Fireflies inhabit ecologically diverse habitats, including wetlands, grasslands, forests, agricultural fields, and urban parks [10]. Firefly population density is correlated with the availability of healthy habitats, and thus they are considered to be reliable indicators of environmental health [17,18]. However, fireflies now face various threats, such as light pollution, habitat loss, and extreme climate change [10,19,20,21,22,23,24]. Thus, there is a strong demand in firefly conservation for a diagnostic tool that evaluates the population densities and behavioral changes in wild fireflies.

For a field researcher, identifying a firefly species via its luminous signaling patterns in the wild still relies heavily on personal experience. Early studies of luminous signals and the nightly behaviors of wild fireflies used direct identification, visual cues, or simply recording systems, such as a photo-sensor (photomultiplier tube) or a stop-watch [13,21,25,26]. However, these methods are unable to record multiple firefly signals occurring simultaneously, such as population flash activities. Recently, improvements in photosensitivity and digital video recording technologies have allowed the luminous signaling to be recorded and stored as digital information. Hence, the spatiotemporal image analysis of flash patterns from a single individual or a population can be directly processed using a personal computer [27,28]. Flash patterns are thought to be a luminous marker for firefly species identification and classification, but this theory has only been investigated in a few species, such as members of the genus *Photinus* [13,14]. Both flash interval (FI) and flash duration (FD) are the key timing characteristics of firefly flash patterns, usually for evaluating changes in flash behaviors under certain conditions, such as courtship and temperature [29,30,31]. No previous study has definitively determined whether FI and FD are specific enough to use in species identification.

About 56 firefly species have been reported in Taiwan [32,33,34]. Among them, *Abscondita cerata* (previously named *Luciola cerata*) is the most abundant. It is widely distributed from sea level to an elevation of 1500 m [34,35,36,37]. The mating season of adult *A. cerata* occurs from March to May. Adult *A. cerata* are active in various habitats, including densely vegetated sloping fields, forest trails, and unlit roads [37]. Their habitats are mostly characterized by high humidity and a lack of light pollution [37]. During their mating season, *A. cerata* are often found to cohabit with more than one other firefly species [37,38]. However, how they coexist without interfering with each other is poorly understood. A nightly census with video-imagery identification is assumed the best way to understand the nocturnal behaviors of *A. cerata* and other cohabiting species. In this context, we initially surveyed the luminous features (FI and FD patterns, flash colors) of three sympatric Taiwanese fireflies (*A. cerata*, *Luciola kagiana*, and *Luciola curtithorax*) to discover the species-specific luminous marker. We further investigated whether any species-specific luminous marker can be applied to assess the nightly changes in species compositions and flash behaviors of a wild sympatric population during *A. cerata* massive occurrence.

## 2. Materials and Methods

### 2.1. Study Habitat

All field studies were conducted on the Genliao hiking trail (25°01′55″ N 121°38′29″ E) in the Nangang mountain system of Taipei City from April to May, 2017 to 2019. The trail is ~1.5 m wide and ~2.5 km long, and surrounded by bushes and trees on both sides (Figure 1A). The trail section (~100 m long) with the most firefly activity was selected for this study. The study section was about 170 m from the nearest artificial road light and shaded by trees without any light pollution. The studied place is not private property or a natural preservation zone (https://conservation.forest.gov.tw/reserve (accessed on 20 November 2021)), and no specific permit was required for the described field studies. The environmental temperature and relative humidity of the habitat were collected by a HOBO data loggers (U12-012, Onset Computer Corp., Bourne, MA, USA), placed at the trailside in the studied region. The collected information was stored in ASCII format for further analysis.

### 2.2. Wild Flash Signal Collection

Flash signals were captured using a digital video camera (FDR-AXP55 4K Handycam^®^ with Exmor R™ CMOS sensor, Sony Corp., Tokyo, Japan). The camera was set to night shot mode and standard steady shot with manually focusing at infinity. While shooting, all artificial lighting on the camera (e.g., screen light) was turned off or shaded with aluminum foil. The shooting was performed at a frame rate of 30 or 60 fps. The recordings were then saved to MPEG-4 or MTS formats. To establish a flash pattern database, flash signals lasting about 20–60 s were shot via handheld tracking on a flying or perching firefly (see examples in Appendix A). During the tracking, the camera was kept about 1–2 m from the target firefly. The target specimen was later collected for further species/sexual identification and flash color measurement. To document the population flash activities, the flashes lasting about 2 min were recorded every 15–60 min from sunset to midnight with fixed-point camera shooting. The camera with an attached tripod stand (~42 cm from the ground to the bottom of the camera) was placed at a fixed point on the side of the trail for shooting.

### 2.3. Video-Imagery Processing and Analyzing of Flash Signals

The videos in MPEG-4 or MTS format were converted into analyzable frame sequences in Tagged Image File (TIF) format using the free software, Video to Image Converter (FreeSTUDIO, DVDVideoSoft, v5.0.101 build 201, Digital Wave ltd., London, UK). All converted frame sequences with a resolution of 3840 × 2160 pixels per frame had a 0.03 or 0.02 s frame delay. The numbers of frames required for analysis were initially evaluated. For instance, to analyze a video clip lasting 5 s at 30 fps, total of 150 frames were required. The frames were then imported as a virtual stack in the free image analyzing software FIJI ImageJ 64 bits (1.53c version, NIH Bethesda, MD, USA) [39] via the toolbar ’File/Import/Image sequence’. To obtain the average intensity of three primary colors (red, yellow, blue) from firefly flashing or the background, all imported frames were converted to monochrome format via the toolbar ‘Image/Type/8-bit grayscale’. Time-stacking images were obtained by superimposing the frames via setting the range of projected frames and the max intensity of spots in the toolbar ‘Image/stacks/Zproject’ or ‘Image/Hyperstacks/Temporal-Color Code’ (Appendix A). To construct a timing diagram of the flash pattern, the flash spots from an individual firefly were manually digitized, one by one. Initially, the appropriate size of the region of interest (ROI) was assigned to the targeted flash spots in virtual stacking frames using the cursor. The selected ROI was then converted into a timing diagram to show the intensity change in ROI within the analyzed frame ranging via the toolbar ’Image/stacks/plot Z-axis profile’. The intensity data were then saved in the comma-separated values’ (CSV) format file via the toolbar ’Data/save data’ in a previewed micrograph. The X–Y coordinates (units in pixels) of each flash spot were obtained by pointing the cursor to the center of each flash in virtual stacking frames. The X–Y coordinates displayed on the toolbar were then recorded to measure their pixel distance. The pixel distance between the two closest flashes was then calculated using a free online 2D distance calculator (https://www.calculatorsoup.com/calculators/geometry-plane/distance-two-points.php (accessed on 20 November 2021), CalculatorSoup LLC., Ashland, MA, USA).

Automated flash spot detection was conducted using the FIJI/ImageJ 64 bits (1.53c version, NIH Bethesda, MD, USA) installed with ‘FFmepg plugin’ and ‘Find Maximal plugin’. The video clip in MPEG-4 format was imported to the FIJI/ImageJ via the toolbar ‘File/Import/movie (FFmpeg)’ and set to ‘use virtual stack’, ‘first frame’ in ‘0′, and ‘last frame’ in ‘–1′. To reduce repeatedly detecting the same flash events in adjacent frames, only one frame was extracted from every 30 frames for flash spot counting. Frame extraction and frame projection were performed by setting the initial ‘reduction factor’ to ‘30′ in the toolbar ‘Image/Stacks/Tools/Reduce’, and then setting the range of slices and the ‘Projection type’ in ‘Max intensity’ via the toolbar ‘Image/Stacks/Zproject’. Automated detection of flash spots in each frame was performed by setting the ’prominence’ to ’80′ and the ‘output type’ to ‘point selection’ in the toolbar ‘Process/Find maximal’. Total detected flashes and their X–Y coordinates were obtained via the toolbar ’Analyze/Measure’, and then were saved in the CSV format for post-hoc examination.

### 2.4. The Measurement and Pattern Matching of FI and FD

To plot a timing diagram of the flash pattern, time-series spot intensity data were loaded into Prism (v6.01, GraphPad Software Inc., CA, USA). Each timing diagram was created by a single profile that displayed multiple pulses per profile, or by overlapping profiles that displayed a single pulse per profile. The FI (or FD) data points were directly measured on the diagram as the definition (iii in Figure 2A–C, magnifying panels). The statistical results of determined FI (or FD) data points were expressed using a boxplot created by OriginR (v6.0, OriginLab Corp., MA, USA). The minimum, maximum, median, first quartile (Q1), and third quartile (Q3) of FI (or FD) data points were obtained directly from the boxplots. The Min–Max range of FI and FD data points for known species (flying males of *A. cerata*, *L. kagiana*, and *L. curtithorax*) was set as the reference range. For each individual of unknown species, the percentages of FI and FD data points matching the reference range were evaluated. The species was identified if over 50% of the total FI (or FD) data points on an individual of an unknown species matched that of a known species, and ≤50% of its total FI (or FD) data points matched those of another known species. Individuals with FI (or FD) data points matching ≤50% of those of any known species were not matched to a species.

### 2.5. Time-Course Specimen Collection

Five specimens in flight and five on perch were collected along the trail every 15 min, or 60 min after sunset, using a sweep net (net frame size: 15 × 12 cm^2^; mesh size: 1 mm). Both specimen collection and fixed-point camera shooting of population flash activity were carried out simultaneously. To avoid interfering with the shooting of flash activity, specimen collections were performed about 20 m behind the camera. The flash patterns and the flash colors of specimens were ignored during collection. Each collected specimen was stored in a 50 mL sampling tube labelled with the collection date and time, place, and flash behaviors (flying or perching) for the following species and sex identifications.

### 2.6. Morphological Identification of Species and Sex

After the specimens were collected from the field, each was immediately transferred and maintained in a 50 mL sampling tube with moist paper tissues in the laboratory, under a 12L:12D photoperiod at 25 °C. Species and sex identification was performed using a stereo microscopes (SMZ745T, Nikon Instruments, Tokyo, Japan) equipped with a digital video camera (FDR-AXP55 4K Handycam^®^ with Exmor R™ CMOS sensor, Sony Corp., Tokyo, Japan). During the survey, the specimen was anesthetized by chilling on ice. *A. cerata*, *L. kagiana*, and *L. curtithorax* were identified based on morphological traits described in previous reports [34,38], including body length (pronotum to the end of abdomen) and unique marks and colors on the elytra, pronotum, and ventral thorax. The sex of specimens was discerned by their light organs (double tagmata for males, single tagma for females) located on the last two abdominal ventrites. All surveys were completed within two days of the collection. One to five identified specimens were sacrificed using CO_2_, and then stored at −20 °C in our lab. All stored specimens were further verified by Dr. Wang Liang-Jong (entomologist). To reduce the impact of specimen collection on the local firefly population, the remaining living specimens were released back into their habitat after identification.

### 2.7. Flash Wavelength Measurement

The emission spectra (color) of flashes were measured using a spectrometer (USB2000+, Ocean Optics Inc., Dunedin, FL, USA). During the measurement, each firefly was immobilized in a homemade firefly holder (Appendix A). The reflectance reading (400–700 nm) was collected from a circular spot (diameter 2 mm) ~3–5 mm below the sample (ventral abdominal light organ) through a glass window of the holder. Five readings from immobilized males were recorded in complete darkness at 25 °C with 75% humidity. The average of the emission spectra readings was calculated for each firefly. All measurements were completed within two days of the specimen collection.

### 2.8. Data Statistics

Significant differences between distinct species, between distinct sexes, and between flying and perching individuals were examined for various characters (FI, FD, flash–flash pixel distance, body size, and flash wavelength) with Student’s *t*-tests. * denotes *p*-value < 0.05, **denotes *p*-value < 0.01, ***denotes *p*-value < 0.001. The ‘*n*’ represents total flash numbers or total data points of FI or FD, while the ‘*N*’ represents total individuals in this study.

## 3. Results

### 3.1. Video-Imagery Distinguished between Flashes from Flying and Perching Individuals

During the mating season (April to May) of *Abscondita cerata*, we noticed that fireflies exhibited two distinct flash behaviors (during flight and while perching) after sunset. These two flash behaviors can be discriminated via a spatiotemporal analysis of a short video clip (5–10 s). The image with time-stacking showed that flying adults produced oblong- (or streamline-) shaped yellowish flashes (right inner panel in Figure 1A), forming a discrete flash trajectory (Figure 1A, blue arrow-lines). Perching adults created round-shaped yellowish flashes (Figure 1A, left inner panel), forming flash clusters at the same or proximate locations (Figure 1A, white hollow circles). The X–Y coordinate analysis of flashes further demonstrated that the flash–flash pixel distances between the flying and perching adults were significantly different (*p*-value < 0.01). The average pixel distance between two of the closest flashes was estimated to be 59.2 ± 27.4 pixels (*n* = 18) for those flying (*N* = 3), and 11.1 ± 5.5 pixels (*n* = 15) for those perching (*N* = 5). Six flying (F1, F2, and F3) and perching (V1, V2, and V3) individuals were randomly selected to further analyze their flash patterns. Flying and perching adults displayed a significant difference in flash patterns and flash frequencies (Figure 1B,C). Flying individuals had highly consistent flash patterns, with high flash frequency compared to perching individuals. The flash frequencies were 1.2–1.6 flashes/s for those flying (*N* = 3) and only 0.2–0.8 flashes/s for those perching (*N* = 3). All these results indicated that the flying and perching adults could be discriminated by their flash shapes, flash trajectories (or clusters), flash frequencies, and flash–flash pixel distances in the video-imagery analysis.

### 3.2. Discovering a Firefly Species-Specific Luminous Marker for Distinguishing Sympatric A. cerata, L. kagiana, and L. curtithorax

The Genliao hiking trail (the studied habitat) contains not only the dominant species, *A. cerata*, but also two other sympatric species—*Luciola kagiana* and *Luciola curtithorax*. The three species have their own unique flash patterns (see Appendix A). We recorded continuous flash signals lasting ~20–30 s from each individual in flight or on perch from 18:30 to 04:00 using a digital camera. The flash-recorded specimens were then harvested for subsequent morphological species/sexual identification and flash wavelength measurement. A comparative graph clearly showed the difference in morphological and luminous features of adult males between these three species (Figure 2). Adult males of both *A. cerata* and *L. kagiana* share highly similar morphological traits (e.g., body sizes, elytra colors), except for the colors of the pronotum and ventral thorax (i in Figure 2A,B). The pronotum is pinkish for adult male *L. kagiana*, but orange-yellowish for adult male *A. cerata*. Unlike *A. cerata*, both *L. kagiana* and *L. curtithorax* possess a unique dark red color on their ventral thorax (i in Figure 2B,C, arrows). Compared to the other two species, the adult male *L. curtithorax* has a relatively tiny body and a unique irregular dark mark on its orange-yellowish pronotum (i in Figure 2C, arrow). The average body size (length from the pronotum to the end of the abdomen) is about 8.49 ± 0.33 mm (*N* = 10) for an adult male *A. cerata*, about 9.35 ± 0.78 mm (*N* = 7) for an adult male *L. kagiana*, and nearly 6.17 ± 0.39 mm (*N* = 5) for an adult male *L. curtithorax*. The images with time-stacking revealed a clearly spatiotemporal difference in flash patterns among all three species in flight (ii in Figure 2A–C). Adult male *A. cerata* exhibited a discrete single-pulse flash trajectory with a low frequency (~1.2 flashes/s, *N* = 10). Adult male *L. kagiana* produced a discrete triple-pulse flash trajectory with a middle frequency (~2.2 flashes/s, *N* = 7). Adult male *L. curtithorax* displayed a dense single-pulse flash trajectory with a high frequency (~4.4 flashes/s, *N* = 5).

To determine FI and FD, the images of flash signals were converted into quantitative timing diagrams (iii in Figure 2A–C). The FI is defined as the time length between the peaks of single-pulses (*A. cerata* and *L. curtithorax*) or between the first peaks of adjacent triple-pulses (*L. kagiana*). FD is defined as the time length during the intensity at about 10% the height of a single-pulse signal (*A. cerata* and *L. curtithorax*) or of a triple-pulse signal (*L. kagiana*). Due to the signal resolution limitation, adjacent triple-pulses of *L. kagiana* are considered as a single flash signal. We performed a statistical analysis on a total of 129 of FI data points and 144 of FD from ten adult males of *A. cerata*, 102 of FI and 109 of FD from seven adult males of *L. kagiana*, and 91 of FI and 94 of FD from five adult males of *L. curtithorax* (Table 1).

The comparative boxplots showed clear differences in FI and FD patterns in adult males among these three species (Figure 2D,E). We found that the Min–Max range of FI (comprising 90% FI data points) was highly species-specific, displaying significant differences (*p*-value < 0.01) among the three species. The FI Min–Max ranges were 0.6–1.2 s for male *A. cerata*, 0.42–0.53 s for male *L. kagiana*, and 0.07–0.43 s for male *L. curtithorax* (Table 1a). In contrast, significant differences in the Min–Max range of FD (comprising 90% FD data points) were found between male *A. cerata* and male *L. kagiana* (*p*-value < 0.01) and between male *A. cerata* and male *L. curtithorax* (*p*-value < 0.01), but not between male *L. kagiana* and male *L. curtithorax* (*p*-value = 0.332). The FD Min–Max ranges were 0.1–0.29 s for male *A. cerata*, 0.07–0.15 s for male *L. kagiana*, and 0.03–0.2 s for male *L. curtithorax* (Table 1b). No flash data from flying females were collected during the study period. Nevertheless, all these results indicated that FI patterns (Min–Max range) can act as luminous markers that effectively discern among the flying males of these three species better than the FD patterns.

In addition, significant differences in flash color (or spectrum) were found only between *Abscondita* and *Luciola* (Figure 2F). The peak of the spectrum appears at 563 nm (yellowish) with a FWHM (full width at half maximum) of 59 nm for adult male *A. cerata* (*N* = 5); at 574 nm (yellowish-orange) with a FWHM of 48 nm for adult male *L. kagiana* (*N* = 5); and at 571 nm (yellowish-orange) with a FWHM of 53 nm for adult male *L. curtithorax* (*N* = 5). λ*_max_* could only be used to distinguish between male *A. cerata* and male *L. kagiana* (*p*-value < 0.01) or between male *A. cerata* and male *L. curtithorax* (*p*-value < 0.01), but not between male *L. kagiana* or male *L. curtithorax* (*p*-value = 0.0*77*).

The flash patterns while perching were also investigated. However, only the flash signals of perching *A. cerata* were successfully collected. Our comparative timing diagram clearly showed a difference in flash patterns between flying and perching *A. cerata* (Figure 3A). Unlike the flying males, the perching males generally emitted low, frequent flash signals (~0.4 flashes/s, *N* = 11). Moreover, the perching females displayed unique flash signals that were a mix of high and low frequencies, different from the mating signal (a long duration flash) of female *A. cerata* [37,38]. The comparative boxplots showed the differences in FI and FD patterns between flying and perching (Figure 3B,C). We statistically analyzed 119 data points of FI and 130 of FD from eleven perching males, and 35 of FI and 38 of FD from three perching females (Table 2), and then compared them with those of flying males. The FI Min–Max ranges were estimated at about 0.53–4.83 s for perching males and 0.6–2.97 s for perching females. The FD Min–Max ranges were 0.07–0.4 s for perching males and 0.1–0.3 s for perching females. The FI pattern demonstrated a highly significant difference between flying and perching males (*p*-value < 0.001) and between perching males and females (*p*-value < 0.001), but only a slightly significant difference between flying males and perching females (*p*-value = 0.03). In addition, the FD pattern showed a highly significant difference between flying males and perching females (*p*-value < 0.001) or between perching males and females (*p*-value < 0.001), but a slightly significant difference between flying and perching males (*p*-value = 0.014).

### 3.3. Evaluating Population Flash Activities during the A. cerata Mating Season

To collect population flash signals, a fixed-point digital camera was set up along the Genliao hiking trail. The shooting was performed every 15 or 60 min from 18:30 to 24:00. A total of 53 video clips (about 2–3 min/clip) were obtained throughout the 2018 mating season: at the beginning (13 April), during the massive occurrence (21 April), and at the end (5 May). Population flash activity (total flashes/s) was initially analyzed using an automatic flash spot detection method, as described in the Materials and Methods. This method can automatically detect and count the total number of flash spots per frame. Time-course analyses of population flash activities clearly revealed that the studied fireflies had two active periods: 18:30–21:30 and 21:30–24:00 (see Appendix A). During the first period (18:30–21:30), the highest flash activity detected was 2.52 flashes/s on 13 April, 1.84 flashes/s on 21 April, and 0.61 flashes/s on 5 May. After 21:30, the highest flash activity detected was 0.2 flashes/s on April 13, 0.25 flashes/s on 21 April, and 0.65 flashes/s on 5 May. These results indicated that there might have been two distinct firefly groups in the habitat, active during different nighttime periods, which had different massive occurrence times during the same season. However, automatic flash spot detection could not provide any details regarding the flash patterns or species compositions for these two firefly groups.

We selected the night when *A. cerata* occurrence was highest (21 April 2018) for our case study. The images with time-stacking initially revealed a nightly change in flash activities and flash patterns (Figure 4A). At 18:30, only a few flash spots occurred in the dark regions of bushes. At 19:00, flash spots clearly increased, spreading along two sides of the trail. At 20:43 and 22:16, flash spots were dramatically lower. For 18:30–20:43, flash trajectories with single-pulse patterns (Figure 4A) that were very similar to those of male *A. cerata* were detected (Figure 2A). At 22:16, there was a flash trajectory with a triple-pulse pattern (Figure 4A) that was very close to that of male *L. kagiana* (Figure 2B). This survey did not find any flash pattern that was similar to that of male *L. curtithorax*. Individuals with similar single- and triple-pulse patterns were consistently observed during the same nightly period on 5 May, but not on 13 April (unpublished data). All the above results indicated that male *A. cerata* was dominant before 22:16 and male *L. kagiana* after.

A detailed quantitative survey of nocturnal activities was conducted on 21 April 2018. A total of 19 video clips (recorded 18:30–23:00) were analyzed. As per the described method (Figure 1), a total of 265 fireflies and their flashes (total 1578 flashes) were discovered in these clips with manually tracking, including 104 flying individuals and 161 perching individuals (see Appendix A). The nightly changes in flash activities (total flashes/s) were highly consistent with the changes in the numbers of fireflies (Figure 4B). The nightly changes in the numbers of fireflies in flight and on perch were also investigated (Figure 4C). At 18:45–19:15, there were more flying than perching fireflies, by about a 1.3:1 ratio. After 19:15, the number of perching individuals was greater than that of flying ones; this ratio could reach 7:1 at one point. Unlike those perching (gray bars), the flying population (black bars) was apparently composed of two groups that were active during distinct time-periods (18:30–21:13 and 22:16–22:44).

### 3.4. Species Identification for the Flying Population Using FI and FD Pattern Matching

FI and FD pattern matching were used to better understand the species compositions of the two flying groups (Figure 4C, black bars). The species identification was conducted by matching the FI (or FD) pattern of each unknown individual with the references (Min–Max ranges of FI or FD) of known species (Figure 2E,F). In this analysis, we only focused on individuals that produced at least seven flashes in 10 s—about 75% (78/104) of all flying individuals (see Appendix A). A total of 847 FI and 923 FD data points were obtained from these individuals. Based on the FI patterns, these flying individuals were divided into three groups (G1–G3). To simplify the results, a small-scale analysis of 15 individuals that were randomly selected from G1 to G3 showed clear differences in FI and FD patterns between these three groups (Figure 5A,B). The FI patterns in G1 and G2 correspond strongly to *A. cerata* (gray bar) and *L. kagiana* (orange bar), respectively (Figure 5A). No FI pattern similar to *L. curtithorax* (red bar) was found in G1–G3. In contrast, the FD patterns in G1 and G2 partially matched *A. cerata* (gray bar) and *L. kagiana* or *L. curtithorax* (orange bar), respectively (Figure 5B). For those in G3, their FI (or FD) patterns did not match *A. cerata*, *L. kagiana*, or *L. curtithorax*.

The species matching percentages of 78 individuals in G1–G3 were extensively evaluated. We identified species based on whether the studied individual had >50% of FI (or FD) data points matched to one of the known species (*A. cerata*, *L. kagiana*, or *L. curtithorax*), and also had ≤50% of its FI (or FD) matched to the other two species. We considered an individual to be unmatched if its FI (or FD) data points matched those of the other known species by ≤50%. A total of 56 individuals in G1 with their FI (*n* = 563) and FD (*n* = 620) data points, 10 individuals in G2 with their FI (*n* = 156) and FD (*n* = 165), and 12 individuals in G3 with their FI (*n* = 128) and FD (*n* = 138) were analyzed.

The range of FI data points per individual was 0.23–2.5 s in G1, 0.33–1.53 s in G2, and 0.13–2.03 s in G3. For each individual in G1, 53–100% of the total FI data points matched those of *A. cerata*, and less than 43% matched those of *L. kagiana* or *L. curtithorax*. For each individual in G2, 56–100% of the total FI data points matched those of *L. kagiana*, and less than 34% matched those of *A. cerata* or *L. curtithorax*. For each individual in G3, ≤50% of the total FI data points matched those of known species. Based on this analysis, the studied population comprised 72% (56/78) male *A. cerata*, 13% (10/78) male *L. kagiana*, and 15% (12/78) unmatched individuals (Figure 5C).

The range of FD data points per individual was 0.03–0.33 s in G1, 0.03–0.23 s in G2, and 0.07–0.3 s in G3. In G1, 25 individuals had 75–100% of their total FD data points matching that of *A. cerata*, and ≤50% of the total FD data points matching that of *L. kagiana* or *L. curtithorax*. In G2, only one individual had 100% of its total FD data points matching that of *L. kagiana* or *L. curtithorax* and had 45% of the total FD data points matching that of *A. cerata*. Three individuals in G2 had 100% of their total FD data points matching that of *A. cerata* and <42% of the total FD matching that of *L. kagiana* or *L. curtithorax*. In G3, three individuals had 92–100% of their total FD data points matching that of *A. cerata* and less than 39% of the total FD data points matching that of *L. kagiana* or *L. curtithorax*. Thus, the FD matching results suggested that the flying population comprised 40% (31/78) of male *A. cerata*, 1% (1/78) of male *L. kagiana*, and 59% (46/78) of unmatched individuals (Figure 5D).

The above results indicated that FI matching had about 85% accuracy in identifying species in the flying population, far higher than the 41% from FD matching. In addition, we estimated that only 33% (26/78) of the studied individuals had both FI and FD patterns (>50% of the total FI or FD) matching to either male *A. cerata* or male *L. kagiana*.

### 3.5. Assessing Nocturnal Active-Periods of FI- and FD-Identified Fireflies

Time-course analysis of FI-identified flying fireflies demonstrated that all *A. cerata* were active from 18:45 to 21:13, and *L. kagiana* were sporadically active from 18:45 to 22:44 (Figure 5E). A similar observation was also made for *A. cerata* based on the analysis of FD-identified populations (Figure 5F). From 18:45–21:13, 1–13 FI- and 1–7 FD-identified *A. cerata* were detected every 10 s. *A. cerata* was most active at 19:00, when 14 FI- and 6 FD-identified individuals were detected per 10 s. In contrast, three or fewer FI- and FD-identified *L. kagiana* were sporadically detected every 10 s from 18:45 to 22:44. After 22:13, *L. kagiana* became the only active species, with two or fewer FI- or FD-identified individuals detected per 10 s. These results indicated that the nightly active period (~2.5 h) of flying *A. cerata* is apparently shorter than that (~5 h) of flying *L. kagiana*, and that they partially overlapped.

To verify the species compositions and active-periods determined by FI- and FD-identification, time-course specimen collection and morphological species identification of fireflies were carried out. Five flying and five perching specimens, ignoring their flash patterns and flash colors, were randomly sampled every 15 min (before midnight) or 60 min (after midnight) from 18:37 on 21 April to 05:02 on 22 April 2018. The active period determined by specimen collection was highly consistent with that determined by FI- and FD-identification (Figure 6A). During 18:37–21:20, flying male *A. cerata* was the dominant species, comprising three to five (60–100%) of the harvested specimens per sampling time. After 21:20, two (40%) or fewer flying male *A. cerata* were sporadically collected per sampling time. In contrast, flying male *L. kagiana* was the minor species before 21:40, and two (40%) or fewer specimens were sporadically harvested per sampling time. During 21:40–01:05, flying male *L. kagiana* became the dominant species, comprising three to five (60–100%) of the collected specimens per sampling time. After 01:05, two (40%) or fewer flying male *L. kagiana* were sporadically collected per sampling time. At 05:02 (dawn), no more flying fireflies were found. All the above results indicated that 21:13–21:20 is the critical time-period that determines the nightly activities of flying males of *A. cerata* and *L. kagiana*.

Among all perching specimens (Figure 6B), male *A. cerata* had the largest population. Perching male *A. cerata* were active throughout the entire night (18:37–05:02) and comprised one to five (20–100%) of the collected specimens per sampling time. Before 23:16, female *A. cerata* (in flight and on perch) were sporadically collected, comprising only one (20%) or no specimens per sampling time. After 23:16, female *A. cerata* (in flight and on perch) became highly active, comprising up to four (80%) of the collected specimens per sampling time. In contrast, perching male *L. kagiana* were rarely found during the collection time, comprising only one (20%) or no specimens per sampling time. The collections also showed that female *L. kagiana* (in flight and on perch) were active before 21:58, comprising up to two (40%) of the collected specimens per sampling time. In this survey, only one male *L. curtithorax* in flight was harvested at 19:08. The above results clearly show that the species compositions and active periods of the perching population were completely different from those of the flying populations at night.

## 4. Discussions

### 4.1. Discovering a Species-Specificity Luminescent Marker-FI Pattern by Comparing Inter-Species Luminescent Characters

Flash patterns (e.g., pulse-pattern, frequency) and flash colors are key luminescent markers for identifying species of adult fireflies [3,40]. However, in a wild environment filled with complicated firefly flash signals, identifying the species within a sympatric firefly population via these luminescent markers is a daunting task. Here, we demonstrated how to use video-imagery to distinguish, isolate, and analyze the flash signals of a sympatric population while flying and perching. Through comparing various luminescent features, we found that the FI pattern is an excellent species-specific luminescent marker for distinguishing three adult male species (*A. cerata*, *L. kagiana*, and *L. curtithorax*) while flying. By combining FI pattern matching and specimen collection, both species content and nocturnal behaviors (flashing in flight or on perch, active-periods) of the sympatric fireflies were successfully evaluated in the wild.

To discover species-specific luminescent markers, a comparative database of morphological and luminescent features is required (Figure 2). Using both time-stack imaging and timing diagrams, the differences in flash patterns between flying adult males of *A. cerata*, *L. kagiana*, and *L. curtithorax* were unveiled, including single- or triple-pulse patterns and FI or FD patterns. However, time-stack imaging has visual limitations for recognizing flash patterns. For instance, it was difficult to distinguish triple-pulse patterns of *L. kagiana* and single-pulse patterns of *A. cerata* while they were in slow flight or too far from the camera. Unlike time-stack imaging, timing diagrams provide quantitative parameters of flashes (FI and FD) for assessing species similarities. Compared to the FD, the FI patterns were better at discriminating among adult males of the three species studied. In contrast, FD patterns and flash colors can discern the individuals only at the genus level (*Abscondita* and *Luciola*) (Figure 2E,F). Moreover, intra-species analysis of *A. cerata* showed that both FI and FD patterns can also distinguish between males and females in flight or on perch (Figure 3).

### 4.2. Application of FI Patterns to Evaluate Species Compositions and Nightly Activities of a Sympatric Firefly Population

As far as we know, the synchronous flashing has never been reported in any Taiwanese firefly species. Unlike synchronous flashing, non-synchronous flashing of a sympatric population involves highly complicated spatial–temporal luminous signals, causing difficulties during video-imagery analysis. We found that the projected flash signal images can discriminate between flying and perching fireflies based on their unique visual characteristics (e.g., flash shapes, flash trajectories or clusters) (Figure 1). This finding allowed us to reduce the signal complexities caused by non-synchronous flashing during video-imagery analysis.

Our case study of *A. cerata* massively occurring demonstrated that it is feasible to evaluate the species compositions and nightly behaviors of a sympatric population in the wild using firefly flash timing features. Two species (*A. cerata* and *L. kagiana*) were successfully identified by time-stack imaging and by FI and FD pattern matching (Figure 4A, Figure 5C,D). Compared with the time-stack imaging and FD pattern matching, FI pattern matching identified more species in the studied population. Moreover, FI pattern matching also provided a highly consistent outcome (species compositions of flying males) with that of the specimen collection (Figure 6A). This result strongly supports that the FI pattern matching can be a reliable species diagnostic tool, along with traditional specimen collection methods. However, among the studied population, 15% and 59% of individuals could not be identified by their FI and FD patterns, respectively. These unidentified individuals displayed irregular scattering patterns of FI and FD that did not match those of the three known species. These abnormal FI and FD patterns might be caused by the change in flying angles during patrolling flight or because parts of flashes were obscured by vegetation during camera shooting. It is also possible that these unidentified individuals might, under certain circumstances, such as during male–female communication [29], become stressed by predation [3,41]. Moreover, firefly flash signaling might also be influenced by various environmental factors, such as wind velocity, quality and intensity of moonlight, temperature, or humidity [29,30,40,42,43]. The influences of individual status and environmental factors on flash signaling in Taiwanese fireflies remain to be explored.

We found that the flash activities of *A. cerata* and *L. kagiana* were not only different at a night, but also significantly different between months during the mating-season (see Appendix A). This implies that their mating seasons partially overlap, but the massive occurrence of *L. kagiana* might be later than that of *A. cerata*. The time-course census allowed us to better understand the nightly activities of these two species. Both the results of FI identification and specimen collection clearly indicated that the flying males of *A. cerata* and *L. kagiana* have distinct nightly active periods (Figure 4, Figure 5 and Figure 6). Being active during distinct time-periods might be an evolutionary strategy for preventing cross-species mating. Like *Photuris* fireflies [40,44], the males of *A. cerata* and *L. kagiana* flash in flight to vigorously advertise themselves and search for the responses from perching females. Hence, the mating of *A. cerata* or *L. kagiana* should occur when flying males are highly active. However, the specimen collection showed that rare perching females (*A. cerata* and *L. kagiana*) flashed when flying males were highly active (Figure 6B), suggesting that the females might only respond to specific males. Interestingly, we also noticed that female *A. cerata* (in flight and on perch) became highly active after midnight, when flying males had low activity (Figure 6B). It is possible that the females were looking for a perching place to rest or lay eggs before dawn.

Among the three species studied, *A. cerata* is the best-studied for its mating behaviors [36,38]. Adult male *A. cerata* is thought to be a primary signaler, like many other reported species [3,36,38]. We noticed that adult male *A. cerata* produced regular high-frequency flashes while flying, but demonstrated irregular low-frequency flashes while perching (Figure 3). Thus, flying male *A. cerata* should utilize high-frequency flashes as a primary courtship signal to attract perching females. After the female responded, the males crawled on the vegetation and then flashed at a low-frequency. The above hypothesis is consistent with a previous report [36], showing that perching male *A. cerata* utilized low-frequency flashes (FI ranging: 3.31–5.14 s) to interact with the females. Thus, the low-frequency flashes of males might be a secondary courtship message to request further intimate contact with the female while perching. Unlike the flying ones, perching male *A. cerata* flashed throughout the night (Figure 6B). However, we believe that most perching males were not in their courtship or mating state. Like many other species, adult *A. cerata* do not feed for two weeks of longevity [45]. Hence, energy utilization for flashing should be under strict management. To reduce energy consumption, perching males may generate low-frequency flashes because they are in a resting state. However, why these males do not completely pause their flashing when resting is still unclear. *A. cerata* mating belongs to an HP (*Hotaria parvula*) communication system [36,38]. Hence, *A. cerata* mating choice relies entirely on whether the females are willing to respond to the males. A female *A. cerata* responds to a male’s flashes after about 0.24 s with a unique long-duration flash (FD: ~0.6 s) [38]. During our study period, we also found the flash signal similar to the reported mating response signal of female *A. cerata*, but this was very rare (unpublished data). We found that the perching female *A. cerata* also emitted a complicated mysterious signal (Figure 3A, the lowest pattern), mixing high and low frequencies, different from the reported mating response signal. Using programmable LED flashing electronics to mimic the female’s non-responsive signal (Figure 3A) in the wild might be a way for future studies to understand how other fireflies respond to this mysterious signal.

The mating behaviors of *L. kagiana* and *L. curtithorax* have not been described before. Despite many differences in flash features, adult males of both *L. kagiana* and *A. cerata* share a highly similar pattern in their flash trajectories during long-distance patrol flights (Figure 4A). This suggests that adult male *L. kagiana* might also be the primary signalers, searching for and locating females in a wide area. Unlike *A. cerata* and *L. kagiana*, adult male *L. curtithorax* flashed at high frequencies with weak light intensity (see Appendix A and Figure 2C). According to our observations, male *L. curtithorax* rarely moved long-distances. Hence, the courtship, male–female recognition, and mating of *L. curtithorax* might occur within an extremely dark and small area, such as the deep inside of bushes.

To process and analyze firefly flash signals, various imaging software have been or are being developed, such as TILIA [27,28,46]. Unlike other imaging software, FIJI / ImageJ is highly compatible with almost all computer operation systems and all video-imagery file formats [39]. We demonstrated that FIJI / ImageJ has advantages in processing images and analyzing the flash signals, such as time-stacking, flash spot detecting and tracking, and timing diagram construction. To detect and distinguish the flash signals in the dark, high quality and high-resolution video clips and images are required. However, the longer the digital camera shots, the larger the image data file it will produce. Processing huge image files manually is very time-consuming, especially when identifying every single firefly in a population via their flash signals. Thus, in this study, only short video clips (5–30 s/clip) were analyzed. Improvements are still needed to a) decrease the size of the video and image files without reducing quality and resolution, and b) discern the flash pattern rapidly and automatically. To date, neither FIJI nor TILIA can rapidly and automatically process a huge amount of complicated imaging data, such as luminous signaling from a sympatric population. In recent years, artificial intelligence (AI) on image pattern recognition has been intensively developed for various academic and industrial applications [47,48]. The use of AI in flash signal analysis and firefly species recognition is in development [27], and we anticipate that it will become a future trend. Until then, a complete digital database of firefly species-specific luminous features must be established. The database is necessary for AI to learn how to identify the species-specific flash patterns in the images or videos. Combining the flash pattern database and AI image recognition will become a powerful diagnostic tool for conservation efforts and studies of the nocturnal behavior and habitats of fireflies, especially while firefly tourism has recently become a global recreational activity [49].

## 5. Conclusions

Flash interval (FI) pattern is a more reliable species-specific luminous marker for discriminating *A. cerata*, *L. kagiana*, and *L. curtithorax* based on their flashing in flight than other luminous features (pulse-patterns, flash durations, and flash colors). Through a case study of *A. cerata* massive occurrence (21 April 2018), 78 flying individuals were recognized initially via their flash shapes, flash trajectories, and pulse-patterns in the video recorded from 18:30 to 23:00. Up to 85% of the flying individuals were identified by FI pattern matching. The time-course analysis of the FI-identified populations further revealed that the nightly active period of *A. cerata* (~2.5 h) is significantly shorter than that of *L. kagiana* (~5 h), and that they partially overlap. Thus, like traditional wild specimen collection, FI pattern matching can be a potential diagnostic tool for evaluating the species composition and nocturnal behaviors (flashing in flight and on perch, active-periods) of a sympatric firefly population in the wild.

## Figures and Tables

**Figure 1 biology-11-00058-f001:**
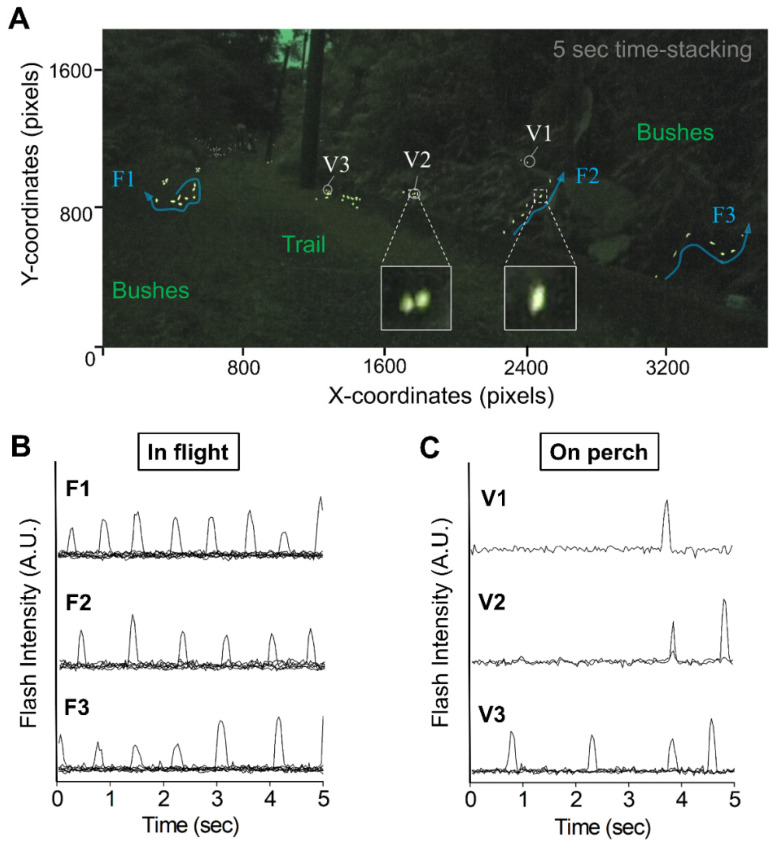
Distinguishing the flash patterns during flight and perching in an *A. cerata* habitat. (**A**) A time-stacking (5 s) graph displays flash activities along the Genliao hiking trail after sunset. Flying individuals (F1–F3) display discrete flash trajectories (blue arrow lines) composed of oblong-shaped flashes (right inner panel). Those perching (V1–V3) exhibit flash clusters (white hollow circles) composed of round-shaped spots (left inner panel). The graph in RGB mode was converted from a 5 s video clip with 30 fps at 18:45 on 21 April 2018. X–Y scales on the left and bottom of the graph denote pixel units. (**B**,**C**) Timing diagrams of flash patterns while flying (F1–F3) and perching (V1–V3). Each diagram comprises a single profile or merged profiles of time-series flash spots’ intensity. A.U: Arbitrary unit. Environmental conditions: Temperature: 22.5 °C; Relative humidity: 80.4%.

**Figure 2 biology-11-00058-f002:**
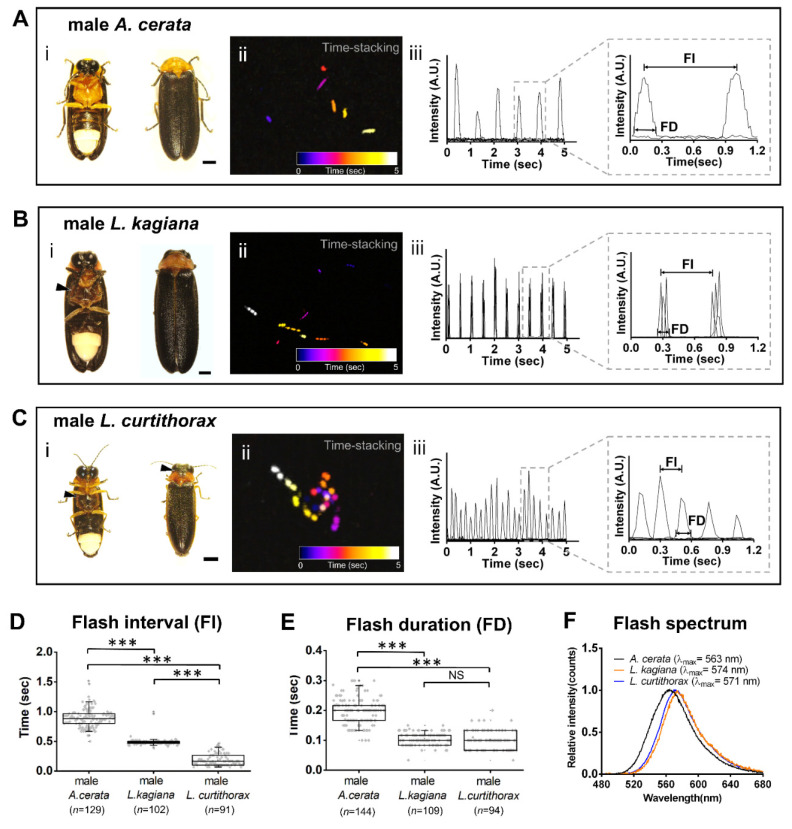
Comparative morphological and luminous features of adult males of *A. cerata*, *L. kagiana*, and *L. curtithorax*. (**A**–**C**) The exteriors and the flash patterns of the three species in flight. Each box gives a (**i**) ventral (left) and dorsal (right) view of the representative individual, (**ii**) a time-stacking (5 s) image and (**iii**) a timing diagram of the flash pattern while flying, and the magnifying panel denotes the measurement of flash interval (FI) and flash duration (FD). In (**i**), the arrows indicate the morphological marks that are distinct from other species. In (**ii**), the pseudo-color flashes correspond to a color code denoting the chronological order of the flashes. (**D**,**E**) Comparative boxplots of FI (or FD) patterns of male adults while flying. The FI (or FD) data points (gray spots) in each boxplot were obtained by continuous flash signals (lasting 5 s) from five to ten distinct individuals. (**F**) Comparative flash wavelength spectra. λ_max_ was determined and averaged from three distinct individuals of each species. ‘*n*’ represents total FI (or FD). Significance: *** *p*  <  0.001. NS: no significance. A.U.: arbitrary units. Scale bar: 1 mm.

**Figure 3 biology-11-00058-f003:**
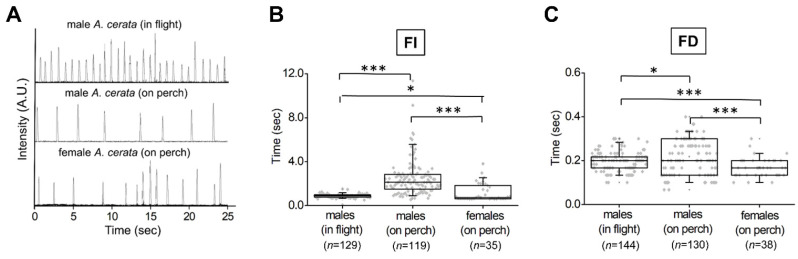
Flash patterns of flying and perching adult males and females of *A. cerata*. (**A**) Comparative flash timing diagrams. (**B**,**C**) Comparative boxplots of FI (or FD) patterns while flying and perching in males and females. The FI (or FD) data points (gray spots) in each boxplot were determined by continuous flash signals (lasting 5–25 s) from 3 to 7 distinct individuals. ‘*n*’ represents the total FI or FD. Significance: * *p*  <  0.05, *** *p*  <  0.001. NS: no significance. A.U.: arbitrary units.

**Figure 4 biology-11-00058-f004:**
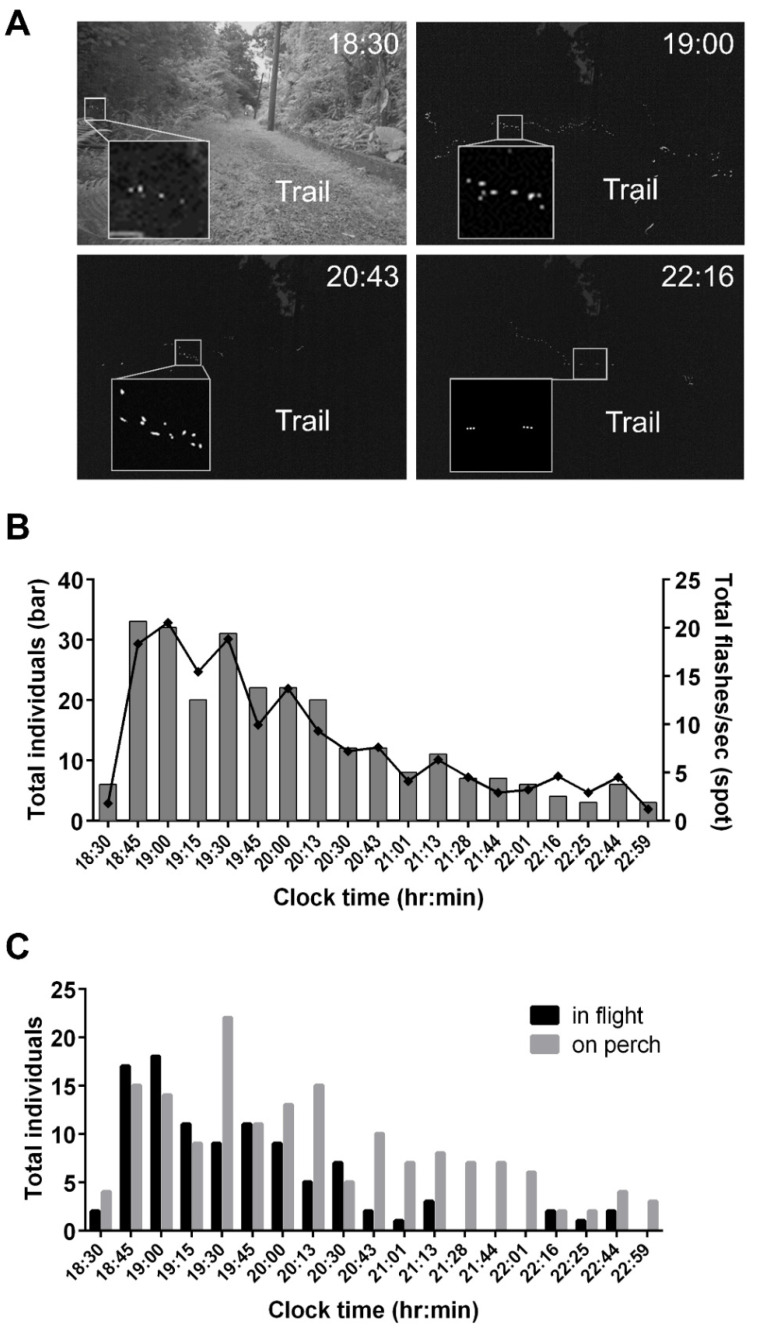
A case study of the nocturnal change in flash patterns and flash activities along the Genliao hiking trail on 21 April 2018. (**A**) Time stacking (10 s) images in grayscale show the change in flash density and flash patterns (inner panels) at 18:30, 19:00, 20:43, and 22:16. (**B**) A time-course bar chart reveals the nightly changes in total flashes/s (black spots) and their corresponding individuals (dark gray bars). (**C**) A time-course bar chart displays the nocturnal change in the number of individuals in flight (black bars) and on perch (gray bars). Both the flashes and individuals in (**B**,**C**) were recognized from 19 video clips (10 s/clip) recorded during 18:30–23:00. The discrimination of flying and perching individuals in the video followed the methods in Figure 1. Environmental conditions during 18:30–23:00: Temperature: 20.7–22.9 °C; Relative humidity: 82.2–93.2%.

**Figure 5 biology-11-00058-f005:**
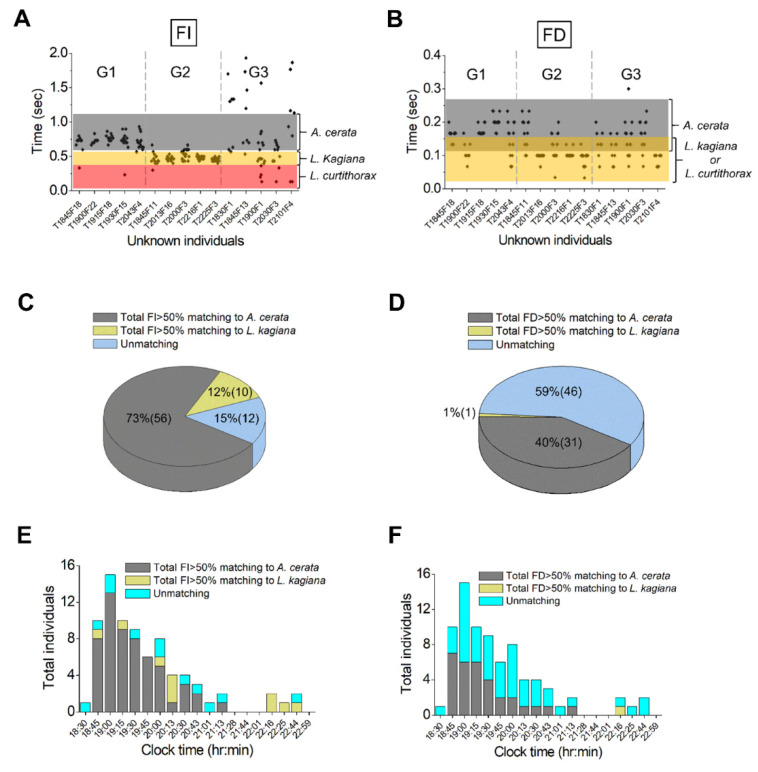
Determining the species composition and active time of the flying population in the case study using FI and FD pattern matching. For the case study on 21 April 2018, flying individuals can be divided into three distinct groups (G1, G2, and G3) according to their FI patterns. (**A**) FI and (**B**) FD patterns of 15 individuals from G1 to G3 are denoted. The FI or FD data points (black spots) of each studied individual (e.g., T1845F18) falling in the reference range of flying males of *A. cerata* (gray bar), *L. kagiana* (orange bar), or *L. curtithorax* (red bar) are shown. Each studied individual contributed 7–21 of the FI data points and 8–22 of the FD data points in this analysis. (**C**,**D**) The pie charts show the percentages and the total numbers (brackets) of *A. cerata* (gray), *L. kagiana* (orange), and unmatched individuals (blue) in the flying population, determined by FI and FD pattern matching. (**E**,**F**) Time-course bar charts display the nightly changes in individual numbers of the FI- or FD-identified *A. cerata* (gray), *L. kagiana* (orange), and unmatched individuals (blue). The details on the FI and FD pattern matching are described in Materials and Methods.

**Figure 6 biology-11-00058-f006:**
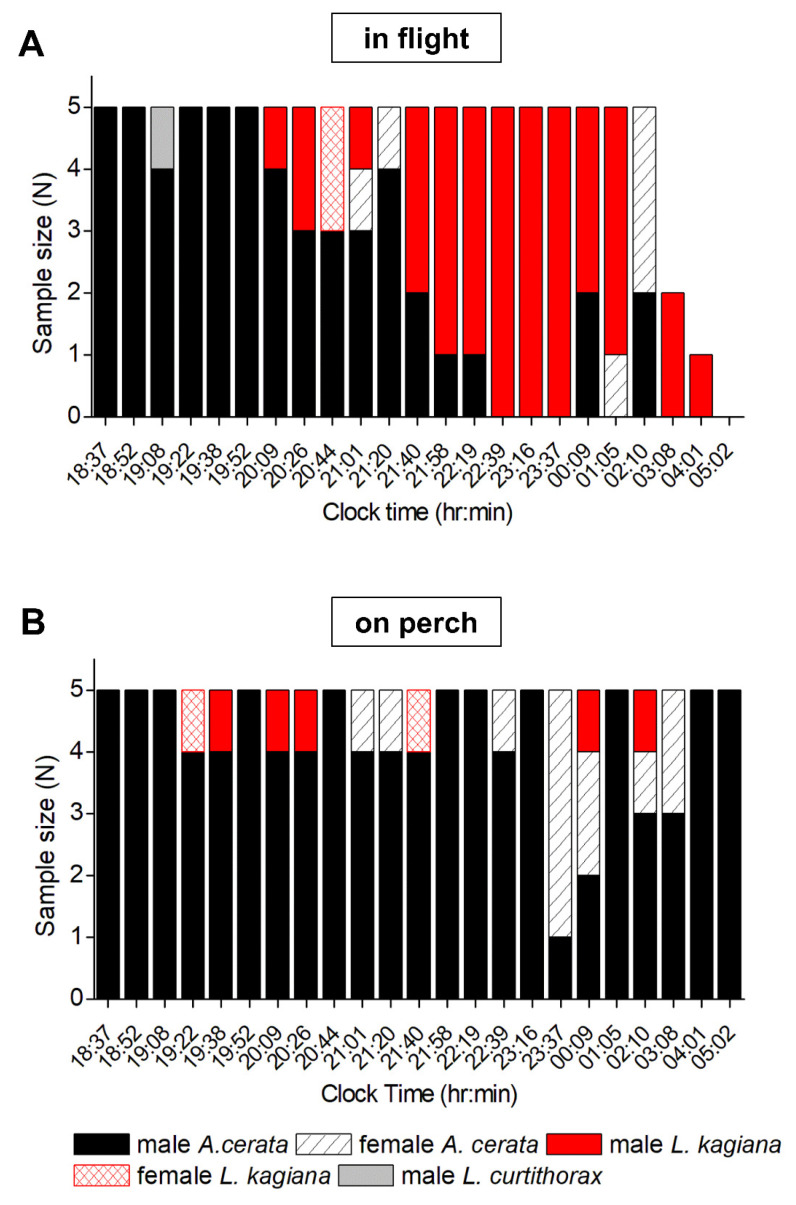
Time-course collection of flying and perching fireflies along the Genliao hiking trail. The nightly harvesting ratio of males or females of *A. cerata*, *L. kagiana*, and *L. curtithorax* while flashing (**A**) in flight and (**B**) on perch are shown. Five specimens in flight and five on a perch were collected every 15 or 60 min from 18:30 on 21 April to 05:01 on 22 April 2018. Morphological identifications of species and sex for collected specimens are described in Materials and Methods.

**Table 1 biology-11-00058-t001:** Statistical results of FI and FD from flying males of *A. cerata*, *L. kagiana* and *L. curtithorax*.

Species	*N*	*n*	Ave Time *	Median *	Q1–Q3*	Min–Max*
(a) FI						
*A. cerata*	10	129	0.89 ± 0.16	0.88	0.8–0.97	0.6–1.2
*L. kagiana*	7	102	0.49 ± 0.09	0.48	0.47–0.5	0.42–0.53
*L. curtithorax*	5	91	0.2 ± 0.14	0.17	0.1–0.27	0.07–0.43
(b) FD						
*A. cerata*	10	144	0.2 ± 0.04	0.2	0.17–0.22	0.1–0.29
*L. kagiana*	7	109	0.11 ± 0.02	0.1	0.08–0.12	0.07–0.15
*L. curtithorax*	5	94	0.1 ± 0.03	0.1	0.07–0.13	0.03–0.2

*N*: total individuals; *n*: total flashes; Ave time: average FI or FD with standard derivation; Q1–Q3: interquartile range of FI or FD (50%); Min–Max: minimum to maximum of FI or FD (90%); *: units in seconds. See also Figure 2D,E.

**Table 2 biology-11-00058-t002:** Statistical results of FI and FD from perching *A. cerata*.

Sex	*N*	*n*	Ave Time *	Median *	Q1–Q3 *	Min–Max *
(a) FI						
Male	11	119	2.48 ± 1.63	2.13	1.4–2.83	0.53–4.83
Female	3	35	1.2 ± 0.81	0.73	0.63–1.83	0.6–2.97
(b) FD						
Male	11	130	0.22 ± 0.08	0.2	0.13–0.3	0.07–0.4
Female	3	38	0.17 ± 0.04	0.17	0.13–0.2	0.1–0.3

*N*: total individuals; *n*: total flashes; Ave time: average FI or FD with standard derivation; Q1–Q3: interquartile range of FI or FD (50%); Min–Max: minimum to maximum of FI or FD (90%); *: units in seconds. See also Figure 3B,C.

## Data Availability

The data presented in this study are openly available in https://drive.google.com/drive/folders/16FDREqsxfIgl3N_lAcFM_Q-8zByjZh8l?usp=sharing (accessed on 20 November 2021).

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
