# Peer review of "Species-Specific Flash Patterns Track the Nocturnal Behavior of Sympatric Taiwanese Fireflies"

_biology, 2022, doi:10.3390/biology11010058_

Round 1

Reviewer 1 Report

I was pleased to review the manuscript by Goh and collaborators entitled “Flash Interval Pattern has Application Potential in Species Identification and Nocturnal Behavior Study for Sympatric Taiwanese Fireflies in the Wild” submitted to the Biology journal. The in vivo bioluminescence emission is a gorgeous and interesting phenomenon, which intrigues researchers all over the world. Even more, when the research topic is associated with intra- and inter-specific communication and the effect of luminosity pollution. In this study, the authors evaluated the use of flash patterns metrics (interval and duration) in three sympatric species to species identification and nocturnal activity behavior evaluation. The work was well designed and performed, with an impressive detailed methodology and results. Usually, as a reviewer, it is common to ask for more details in methodological steps, however here we have more than that. Now, as editor and reader, I believe that we have very extensive “material and methods” and “results” sections, which may be more objective and less stuff. Most of my concern is about English misspelling, which may need need to be revisited before the final acceptance. The authors also described the possible presence of two distinct “groups” that have two distinct high activity periods. I am not completely convinced that. Thus, the authors may bring new evidence or literature that clarifies this point. Another option is to make it pleasant in the text.

My minor comments are:

In general, several words in “tandem” repetitions. For instance, the use of the “firefly” term more than 6 times in the same paragraph. Try to avoid it using pronouns. 

Simple summary:

line 11: replace “lack of” for “a lack of” or “lacking”;

line 12: replace “changes of” for “changes in”

Introduction:

line 51: replace “understand” for “understanding” 

line 78: replace “simple” for “simply” 

line 87: include “genus” before “Photinus”

line 93/94: replace “distribution” for “distributed” 

line 98: replace “is” for “are” 

line 101: replace “cohabit” for “cohabiting”

line 103/104: replace “discovery” for “discover”

ok, mas preciso usar mais pronomes

Several “tandem” repetitions of the “firefly” term. Try to avoid ti using pronouns. 

Material and Methods:

line 141: include “of” before “150 frames”

line 178: include “a” before “single”

line 177/178: chose to keep or remove the hyphen in “multiples-pulses” and “single-pulse”

line 180: include “a” before “boxplot”

line 201: replace “wild” for “field”

line 204: include “with” before “a digital video camera”

line 206: replace “through” for “by”

line 209: replace “were” for “was”

Mto bem detalhado e metódico - de repente pensar em resumir partes

Results:

line 241: replace “difference” for “different”

Figure 1: Really interesting and important, however, its quality, mainly the 1A was too poor. I suggested improving its quality or size. Also, the figure caption presents the wrong number (2 instead of 1); 

line 275: replace “possesses” for “possess”

line 275: replace “its” for “their”

line 305: replace “was” for “were”

line 323: replace “to” for “from”,

line 374: include “of” before “53 video clips”

line 404: remove “of” after “of 19”

line 374: include “of” before “265 fireflies”

line 521: include “The” before “above”

line 523: replace “to” for “from”

Discussion:

line 559: replace “As far as we known, synchronous flashing ...” for “As far as we know, the synchronous flashing...”

line 600: include “were” before “highly”

line 622: replace “belong” for “belongs”

line 627: confuse “Most females emerged a signal (Fig. 3A) different from the response signals.”

line 631: replace “hasn´t” for “haven’t”

line 636: remove “a” from “a weak”

line 639: include “the” before “deep”

Conclusion:

line 667: replace “potent” for “potential”

Author Response

Reviewer 1:

Comments and Suggestions for Authors

I was pleased to review the manuscript by Goh and collaborators entitled “Flash Interval Pattern has Application Potential in Species Identification and Nocturnal Behavior Study for Sympatric Taiwanese Fireflies in the Wild” submitted to the Biology journal. The in vivo bioluminescence emission is a gorgeous and interesting phenomenon, which intrigues researchers all over the world. Even more, when the research topic is associated with intra- and inter-specific communication and the effect of luminosity pollution. In this study, the authors evaluated the use of flash patterns metrics (interval and duration) in three sympatric species to species identification and nocturnal activity behavior evaluation. The work was well designed and performed, with an impressive detailed methodology and results. Usually, as a reviewer, it is common to ask for more details in methodological steps, however here we have more than that. Now, as editor and reader, I believe that we have very extensive “material and methods” and “results” sections, which may be more objective and less stuff. Most of my concern is about English misspelling, which may need need to be revisited before the final acceptance. The authors also described the possible presence of two distinct “groups” that have two distinct high activity periods. I am not completely convinced that. Thus, the authors may bring new evidence or literature that clarifies this point. Another option is to make it pleasant in the text.

Author reply: Very appreciate for your valuable and kindly suggestions. We preliminarily deduce that there were two groups active in different nightly periods during the mating season, based on the time-dependent change of flash numbers (see the paragraph Page 15, Line 401-417). And then, the subsequent experiments (e.g. species identification, specimen collection) have further shown that these two groups belongs to different species and their nightly active periods were different but partially overlapped. Indeed, we still don’t have solid evidence to show that if A. cerata (the dominant species before 21:30) or L. kagiana (the dominant species after 21:30) prefer to courtship or mating during specific night period or during specific period in mating season. Thus, we replaced the ‘were’ to ‘might have’ in the sentence ‘These results indicated that there might have two distinct firefly groups in the habitat, active during different nighttime periods, which had different massive occurrence times during the same season.’ in the Page 15, Line 413.

My minor comments are:

In general, several words in “tandem” repetitions. For instance, the use of the “firefly” term more than 6 times in the same paragraph. Try to avoid it using pronouns. 

Author reply: We have reduced some repetitive words in our manuscript.

  1. Replace ‘firefly populations’ to ‘they’ in Page 2 Line 74
  2. Delete ‘firefly’ in Page 2 Line 85
  3. Delete ‘Firefly’ in Page 2 Line 88
  4. Replace ‘firefly’ to ‘the’ in Page 29 Line 733
  5. Replace ‘firefly’ to ‘the’ in Page 29 Line 734
  6. Replace ‘firefly’ to ‘the’ in Page 29 Line 745
  7. Replace ‘firefly’ to ‘the’ in Page 29 Line 748

Simple summary:

line 11: replace “lack of” for “a lack of” or “lacking”;

Author reply: Already corrected, please check Page 1, Line 12 ..it is still a lack of an effective diagnostic tool to.. ”

line 12: replace “changes of” for “changes in”

 Author reply: Already corrected, please check Page 1, Line 13 … to monitor the changes in their population size and nocturnal..”

Introduction:

line 51: replace “understand” for “understanding” 

Author reply: Already corrected, please check Page 2, Line 54 .. new path towards understanding the molecular..”

line 78: replace “simple” for “simply” 

Author reply: Already corrected, please check Page 2, Line 81-82…visual cues, or simply recording systems, such as a photo-sensor …”

line 87: include “genus” before “Photinus”

Author reply: Already corrected, please check Page 2, Line 90 …such as members of genus Photinus…”

line 93/94: replace “distribution” for “distributed” 

Author reply: Already corrected, please check Page 3, Line 96-97… widely distributed from sea level to…”

line 98: replace “is” for “are” 

Author reply: Already corrected, please check Page 3, Line 101A. cerata are often found to cohabit… ”

line 101: replace “cohabit” for “cohabiting”

Author reply: Already corrected, please check Page 3, Line 104… and other cohabiting species….”

line 103/104: replace “discovery” for “discover”

 Author reply: Already corrected, please check Page 3, Line 107….to discover the species-specific luminous marker….”

Several “tandem” repetitions of the “firefly” term. Try to avoid ti using pronouns. 

Author reply: We have reduce some repetitive words in our manuscript.

  1. Replace ‘firefly populations’ to ‘they’ in Page 2 Line 74
  2. Delete ‘firefly’ in Page 2 Line 85
  3. Delete ‘Firefly’ in Page 2 Line 88
  4. Replace ‘firefly’ to ‘the’ in Page 29 Line 733
  5. Replace ‘firefly’ to ‘the’ in Page 29 Line 734
  6. Replace ‘firefly’ to ‘the’ in Page 29 Line 745
  7. Replace ‘firefly’ to ‘the’ in Page 29 Line 748

Material and Methods:

line 141: include “of” before “150 frames”

Author reply: Already corrected, please check Page 4, Line 149…at 30 fps, total of 150 frames were required…”

line 178: include “a” before “single”

Author reply: Already corrected, please check Page 4, Line 187…that displayed a single-pulse per profile…. ”

line 177/178: chose to keep or remove the hyphen in “multiples-pulses” and “single-pulse”

Author reply: Already corrected, please check Page 4, Line 187…that displayed a single-pulse per profile…”

line 180: include “a” before “boxplot”

Author reply: Already corrected, please check Page 4, Line 189….were expressed using a boxplot created by… ”

line 201: replace “wild” for “field”

Author reply: Already corrected, please check Page 5, Line 212…were collected from field, each specimen…”

line 204: include “with” before “a digital video camera”

Author reply: Already corrected, please check Page 5, Line 215…equipped with a digital video….”

line 206: replace “through” for “by”

Author reply: Already corrected, please check Page 5, Line 217….anesthetized by chilling on ice….”

line 209: replace “were” for “was”

Author reply: please check Page 5, Line 220..The sex of specimens was discerned by….”

Results:

line 241: replace “difference” for “different”

Author reply: Already corrected, please check Page 6, Line 256 adults were significantly different (p-value < 0.01)…”

Figure 1: Really interesting and important, however, its quality, mainly the 1A was too poor. I suggested improving its quality or size. Also, the figure caption presents the wrong number (2 instead of 1); 

Author reply: The quality and size of all figures have been improved, and the figure caption has been corrected. Please check Page 8, Line 270

line 275: replace “possesses” for “possess”

Author reply: Already corrected, please check Page 8, Line 293 ….curtithorax possess unique dark red color….”

line 275: replace “its” for “their”

Author reply: Already corrected, please check Page 8, Line 293 …red color on their ventral thorax…”

line 305: replace “was” for “were”

Author reply: Already corrected, please check Page 9, Line 317 …(comprising 90% FI data points) were highly species-specific….”

line 323: replace “to” for “from”,

Author reply: Already corrected, please check Page 11, Line 344…are distinct from other species…. ”

line 374: include “of” before “53 video clips”

Author reply: Already corrected, please check Page 15, Line 404 ..A total of 53 video clips…”

line 404: remove “of” after “of 19”

Author reply: Already corrected, please check Page 16, Line 434 ..A total of 19 video clips…”

line 374: include “of” before “265 fireflies”

Author reply: Already corrected, please check Page 16, Line 435 …a total of 265 fireflies and…”

line 521: include “The” before “above”

Author reply: Already corrected, please check Page 23, Line 580…was harvested at 19:08. The above… ”

line 523: replace “to” for “from”

Author reply: Already corrected, please check Page 23, Line 582…completely different from those of…. ”

Discussion:

line 559: replace “As far as we known, synchronous flashing ...” for “As far as we know, the synchronous flashing...”

Author reply: Already corrected, please check Page 27, Line 648…As far as we known, the synchronous flashing has…”

line 600: include “were” before “highly”

Author reply: Already corrected, please check Page 27, Line 650…population were highly… ”

line 622: replace “belong” for “belongs”

Author reply: Already corrected, please check Page 28, Line 710 ….mating belongs to an HP (Hotaria parvula)…”

line 627: confuse “Most females emerged a signal (Fig. 3A) different from the response signals.”

Author reply: the description has been changed to“ We found that the perching female A. cerata also emitted a complicated mysterious signal (Fig. 3A, the lowest pattern), mixing high and low frequencies, different from the reported mating response signal.”, please check Page 28, Line 715-718

line 631: replace “hasn´t” for “haven’t”

Author reply: Already corrected, please check Page 29, Line 721L. curtithorax haven’t been described…”

line 636: remove “a” from “a weak”

Author reply: Already corrected, please check Page 29, Line 726 …high frequencies with weak light intensity…”

line 639: include “the” before “deep”

Author reply: Already corrected, please check Page 29, Line 729 …such as the deep inside of bushes…”

Conclusion:

line 667: replace “potent” for “potential”

Author reply: Already corrected, please check Page 29, Line 766… FI pattern matching can be a potential diagnostic tool for…”

Reviewer 2 Report

General remarks

I find this paper very interesting and worth publishing in Biology but there are some errors that should be corrected before publication.

The Authors draw interesting conclusions based on the study of insects. Therefore, they should follow the basic rules related to the morphological nomenclature of this class of arthropods. Morphology is one of the basic fields of entomology. In the case of all insects, the term "body divided into three tagmata" is used - not into "three segments". Tagma and segment are not the same. We distinguish the head, the thorax and the abdomen - there is no such thing as “a tail” in insects. Measurements are made from the front edge of the pronotum to the end of the abdomen. Visible abdominal sternites are called the ventrites. In my opinion, the Authors should necessarily correct morphological nomenclature. It should be assumed that the majority of readers will be entomologists for whom unprofessional nomenclature will significantly reduce the scientific credibility of the text.

Detailed suggestions for corrections are presented below, along with corrections of typical editorial errors.

Detailed comments

Page 3 line 93-94

Is: distribution – should be: distributed

Page 5 line 208

Is: pronotum to tail – should be: pronotum to the end of abdomen

Is: marks; and colors – should be: marks and colors

Page 5 line 210-211

Is: last two body segments – should be: last two abdominal ventrites

Page 5 line 231

The title of the sub-section 3.1.1. is not in italics while all previous sub-sections were. It should be unified throughout the whole text (same with 3.1.2., 3.1.4, 3.1.5)

Page 6 line 241

Is: difference – should be: different

Page 6 figure caption

Is: Figure 2. – should be: Figure 1.

Page 6 line 262

Is: A. cerata, L. kagiana and L. kagiana – should be: A. cerata, L. kagiana and L. curtithorax

Page 7 line 272

Is: except for the colors of their 2nd (pronotum) and 3rd (ventral thorax) body segments (i – should be: except for the colors of the pronotum and ventral thorax (i

Page 7 line 274

Space is missing in A.cerata

Page 7 line 278

Is: (length from 2nd segment to tail end) – should be: (length from pronotum to the end abdomen)

Page 9 table caption line 331

Flying males should not be in italics

Page 18 line 622

Is: is belong to – should be: belongs to

References section – some titles have all words beginning with capital letters, while others do not - this should be unified

Page 22 line 746

Is: phroturis – should be in italics and start with capital letter: Phroturis

Page 23 line 782

Luciola cruciata should be in italics

Page 23 line 800

Is: (Coleoptera : lampyridae) – should be: (Coleoptera: Lampyridae)

Page 23 line 815

Photinus should be in italics

Author Response

Reviewer 2:

General remarks

I find this paper very interesting and worth publishing in Biology but there are some errors that should be corrected before publication.

The Authors draw interesting conclusions based on the study of insects. Therefore, they should follow the basic rules related to the morphological nomenclature of this class of arthropods. Morphology is one of the basic fields of entomology. In the case of all insects, the term "body divided into three tagmata" is used - not into "three segments". Tagma and segment are not the same. We distinguish the head, the thorax and the abdomen - there is no such thing as “a tail” in insects. Measurements are made from the front edge of the pronotum to the end of the abdomen. Visible abdominal sternites are called the ventrites. In my opinion, the Authors should necessarily correct morphological nomenclature. It should be assumed that the majority of readers will be entomologists for whom unprofessional nomenclature will significantly reduce the scientific credibility of the text.

Detailed suggestions for corrections are presented below, along with corrections of typical editorial errors.

Author reply: Thank you so much for your valuable suggestions. We have learned a lot from your advices. We have corrected the issues of morphological nomenclature, e.g. replacing the ‘segment’ to the ‘tagmata’ (Page 5,Line 221).

Detailed comments

Page 3 line 93-94

Is: distribution – should be: distributed

Author reply: Already corrected, please check Page 3, Line 96-97…It is widely distributed from sea level.. ”

Page 5 line 208

Is: pronotum to tail – should be: pronotum to the end of abdomen

Author reply: Already corrected, please check Page 5 Line 219 “…body length (pronotum to the end of abdomen); unique… “

Is: marks; and colors – should be: marks and colors

 Author reply: Already corrected, please check Page 5 Line 220 “ ….marks; and colors on the elytra… “

Page 5 line 210-211

Is: last two body segments – should be: last two abdominal ventrites

 Author reply: Already corrected, please check Page 5 Line 222  “ ..the last two abdominal ventrites …“

Page 5 line 231

The title of the sub-section 3.1.1. is not in italics while all previous sub-sections were. It should be unified throughout the whole text (same with 3.1.2., 3.1.4, 3.1.5)

 Author reply: All titles in italics style of the sub-section (3.1, 3.2…….3.5) have been unified, please check Page 5 Line 246, Page 8 Line 279, Page 15 Line 400, Page 16 Line 444, Page 19 Line 516, Page 23 Line 583, Page 27 Line 645.

Page 6 line 241

Is: difference – should be: different

 Author reply: Already corrected, please check Page 6 Line 256 “…were significantly different (p-value < 0.01). …  “

Page 6 figure caption

Is: Figure 2. – should be: Figure 1.

 Author reply:The figure caption has been corrected. Please check Page 8, Line 270

Page 6 line 262

Is: A. cerata, L. kagiana and L. kagiana – should be: A. cerata, L. kagiana and L. curtithorax

 Author reply: Already corrected, please check Page 8 Line 279 “…for distinguishing sympatric A. cerata, L. kagiana and L. curtithorax…  “

Page 7 line 272

Is: except for the colors of their 2nd (pronotum) and 3rd (ventral thorax) body segments (i – should be: except for the colors of the pronotum and ventral thorax (i

 Author reply: Already corrected, please check Page 8 Line 290-291 “except for the colors of the pronotum and ventral thorax (i in Fig. 2A and 2B)  “

Page 7 line 274

Space is missing in A.cerata

 Author reply: Already corrected, please check Page 8 Line 292 “…Unlike A. cerata, both L. kagiana … “

Page 7 line 278

Is: (length from 2nd segment to tail end) – should be: (length from pronotum to the end abdomen)

 Author reply: Already corrected, please check Page 8-9 Line 296-297 “ …(length from pronotum to the end abdomen) is about …“

Page 9 table caption line 331

Flying males should not be in italics

 Author reply: Already corrected, please check Page 12 Line 351 “…FD of flying males of A. cerata…“

Page 18 line 622

Is: is belong to – should be: belongs to

 Author reply: Already corrected, please check Page 28 Line 710“ ….mating belongs to an HP (Hotaria parvula) communication…. “

References section – some titles have all words beginning with capital letters, while others do not - this should be unified

Author reply: Already unified. 

Page 22 line 746

Is: phroturis – should be in italics and start with capital letter: Phroturis

 Author reply: Already corrected, please check Page 33 Line 842“… mimicry in Photuris fireflies: signal… “

Page 23 line 782

Luciola cruciata should be in italics

 Author reply: Already corrected, please check Page 34 Line 886 “ …variation in the flashes of the firefly Luciola cruciata… “

Page 23 line 800

Is: (Coleoptera : lampyridae) – should be: (Coleoptera: Lampyridae)

Author reply: Already corrected, please check Page 35 Line 915 “…The firefly genus Vesta in Taiwan (Coleoptera: Lampyridae). J Kansas Entomol… “

Page 23 line 815

Photinus should be in italics

Author reply: Already corrected, please check Page 35 Line 932“… Flash signal evolution in Photinus fireflies: character displacement… “

Reviewer 3 Report

  1. The introduction contains a lot of information. However, modern literature about fireflies can be added to this section (https://journals.plos.org/plosone/article?id=10.1371/journal.pone.0191576; https://www.scirp.org/html/4-1270093_53614.htm; https://peerj.com/articles/12121/; https://pubag.nal.usda.gov/catalog/7311659; https://conbio.onlinelibrary.wiley.com/doi/full/10.1111/csp2.391; https://journals.biologists.com/bio/article/10/8/bio058762/271863/Motion-enhancing-signals-and-concealing-cues).
  2. There are some figures of poor quality in the results. It is necessary to improve the quality of the figures.
  3. Line 372-377. Why this description? It should be in the methods.
  4. The authors described the mating behavior and signals of only one species of fireflies. However, their manuscript describes two more types of signals. Please give a more detailed analysis of the other two types.
  5. The authors conducted studies in only one biotope. Why didn't they explore other habitats? Can biotopes influence the signals of fireflies? How can this affect the results?
  6. The conclusion contains very little information. I propose to include in it more specific conclusions obtained by the authors.

Author Response

Reviewer 3:

Comments and Suggestions for Authors

  1. The introduction contains a lot of information. However, modern literature about fireflies can be added to this section (https://journals.plos.org/plosone/article?id=10.1371/journal.pone.0191576; https://www.scirp.org/html/4-1270093_53614.htm; https://peerj.com/articles/12121/; https://pubag.nal.usda.gov/catalog/7311659; https://conbio.onlinelibrary.wiley.com/doi/full/10.1111/csp2.391; https://journals.biologists.com/bio/article/10/8/bio058762/271863/Motion-enhancing-signals-and-concealing-cues).

Author reply: The suggested literatures have been added in the manuscript, please check Page 2 Line 60; Page 2 Line 74; Page 19 Line 671

  1. There are some figures of poor quality in the results. It is necessary to improve the quality of the figures.

Author reply: The quality and size of all figures have been improved.

  1. Line 372-377. Why this description? It should be in the methods.

Author reply: The description is a brief introduction how we designed and performed the experiments, so that the reader could much easy to understand the following results.

  1. The authors described the mating behavior and signals of only one species of fireflies. However, their manuscript describes two more types of signals. Please give a more detailed analysis of the other two types.

Author reply: If not misunderstood, you should be wondering about the mating behaviors and flash signals when perching of other two species (L. kagiana and L. curtithorax). Unfortunately, we only collected one or none of the perching signals of these two species during our study periods. The signal collection is still going on, and might take several years to obtain enough information. As far as we known, no related research regarding the mating and courtship behavior of these two Taiwanese species were formally reported. We could only speculate on their possible mating and courtship behaviors based on our limited results and observations (please see the discussion, Page 29 Line 722-730) 

  1. The authors conducted studies in only one biotope. Why didn't they explore other habitats? Can biotopes influence the signals of fireflies? How can this affect the results?

Author reply: Due to limited resources, we could only focus on a single biome survey. We are planning to expand the scale of surveys on other biotopes in the following years. According to a previous study (Ohba, N.; Yang, P.S. Flash patterns and communication system of the Taiwan firefly, Luciola cerata Olivier. Science Report of the Yokosuka City Museum 2003, 50, 1-12. ), A. cerata’s luminous pattern might have slightly geographically difference from north to south, especially on the flash intervals. The flash intervals of north A. cerata were much shorter than south one, and this data were obtained by six distinct individuals from six different regions. Our results seem to be consistent with the reported flash patterns of nouth A, cerata. However, more detailed investigation and evidence are needed to confirm the influence of the biotope to the signal.

  1. The conclusion contains very little information. I propose to include in it more specific conclusions obtained by the authors.

Author reply: We have added “Through a case study of A. cerata massive occurrence (21 April 2018), 78 flying indi-viduals were recognized initially via their flash shapes, flash trajectories, and pulse-patterns in the video recorded from 18:30–23:00. Up to 85% of the flying individuals were identified by FI pattern matching, but only 41% were identified using FD pattern matching. The time-course analysis of the FI-identified populations further revealed that the nightly active period of A. cerata (~2.5 hr) was significantly shorter than that of L. kagiana (~5 hr), and that they partially overlapped.” in the conclusion. Please check Page 29, Line 761-767 .

Reviewer 4 Report

Manuscript ID: biology-1497730

The paper´s expressed goal is to test the effectiveness of spot video recording and time-stacked analysis of flash behavior to differentiate species and census mixed-species populations in the field. The authors first had to make a flash reference library using time-stacked imagery of individually tracked/video recorded fireflies of three species. Flash interval proved to be a reliable marker for species diagnosis.

Flash interval (FI) patterns seem to show enough reliability (ie low variance) to be species specific, making them suitable for identifying firefly species in the field.

The study indicates that FI patterns may be a reliable species-specific luminous marker for monitoring the daily and seasonal behavioral changes in a sympatric firefly population in the field. Such remote census capability may have applications for firefly conservation.

This is a nice paper.

Documenting flash behavior is a challenge. The original use of photomultipliers and analog-to-digital conversion was inefficient. The value of the present study lies in its use of simple video equipment deployed in the field to document flash behavior in real time.

Major comments:

First, the title is a bit too long. I recommend something like, ¨Species specific flash patterns distinguish sympatric Taiwanese fireflies¨ or "Interflash interval identifies sympatric species in a Taiwanese firefly community.

The paper is too long by 4-5 pages. Much of the results are repetitions of the methods.

There appears to be two Figure 2s. I suppose the first one is actually Figure 1.

I recommend a figure to show the time-stack method like a graphic abstract. It still seems to be a frame-by-frame analysis when it is really a stacked image composed of many separate patterns that become superimposed.

These species seem to be train flashers, ie they produce long sequences of uniform pulses with equal inter-pulse intervals. If so, how do you know when a pulse train begins or ends? It appears to be assumed that the trains are uniform in timing and intensity from beginning to end.

L378: if there are 2 groups, what did your aerial nets reveal? Were samples collected?

The terminology is imprecise and needs to be standardized. It should follow Lewis and Crastley 2008: flash interval, pulse duration, interpulse interval as the main features to describe flash patterns.

I have some trouble with the measurements. For instance, FI is measured from peak intensity rather than from pulse onsets. This is problematic. Onsets are when the female response is triggered. Interestingly, on L399 we see the correct use of the term pulse for the three peaks of intensity this species produces.

I do not see how the authors decided where a flash pulse begins or ends. There is no stated threshold for pulse onset or offset. Thus, interflash intervals, pulse durations and pulse intervals can be quite variable.

It is clear from Figure 2 that cerata and curtithorax are train flashes. Perhaps A. cerata might be considered a single flash species at about 1 hertz. The other two species have flash patterns at 2 and around 7 hertz. However, the intensity diagrams show that L. kagiana actually has a triple-pulsed flash, delivering 3 pulses at 2 hertz! This is too fast for the human eye to detect but the camera captures the peak emissions of each pulse.

So, can one detect the flight path from these stacked images?

L133-134: This is not clear. Was recoding continuous or just for a few minutes?

L215: glad to see this step taken!

L238: what is an arbitrary unit? shouldn't this be relative intensity?

L255: shouldn't this be patrolling flight path or signal flight path? This phase of courtship is generally referred to as "patrolling" when male broadcast their identity as versus "courtship" once a female responds and it located. It appears to be sigmoid.

L353:where is the data for females?

Table 1: the third columns should be labeled Time(s) (X±SD)

Flashes produced by perching males likely have no signal value. They may be trophic flashes, ie those produced while coordinating the complex physiology of the flash-control mechanism. This might be mentioned as reason for the lack of significant differences among species.

I have to question the appropriateness and reliability of the time stacking function. If the objective is to measure the flash pattern, why stack flashes? Why not simply measure the onset/offset of each pulse and the time between pulses?

Further, intensity varies according to the inverse square law whereby intensity falls by the square of distance from the source. Moreover, intensity varies according to the direction of the abdomen during flight. Therefore it is wholly unreliable as a species-specific marker of any kind. However, peak intensity indicates pulse number.

I think the references could better reflect the state of knowledge. For instance, the authors make no mention of the role of sexual selection or predation in shaping flash communication. Variability in flash behavior is the chief means by which females choose mates. This variation bears on the stereotypy of the flash dialogue and thus on the usefulness of flash behavior as a means of species-specific diagnosis. Moreover, courtdships have different phases such as patrol, male-male interactions, female dialoging, and scramble competition-all involving different flash patterns (see Vencl and Carlson 1998).

Minor comments:

Abstract:

L25: change determined to "measured" or perhaps "quantified"

L28: change identifying to "distinguishing" or even "differentiating"

Introduction:

L67: Actually, a great deal more is known about asynchronous species, mainly because there are many, many more asynchronous species, many of which are very common. Correspondingly these have attracted much more attention and we know more about them.

Author Response

Reviewer 4:

Comments and Suggestions for Authors

The paper´s expressed goal is to test the effectiveness of spot video recording and time-stacked analysis of flash behavior to differentiate species and census mixed-species populations in the field. The authors first had to make a flash reference library using time-stacked imagery of individually tracked/video recorded fireflies of three species. Flash interval proved to be a reliable marker for species diagnosis.

Flash interval (FI) patterns seem to show enough reliability (ie low variance) to be species specific, making them suitable for identifying firefly species in the field.

The study indicates that FI patterns may be a reliable species-specific luminous marker for monitoring the daily and seasonal behavioral changes in a sympatric firefly population in the field. Such remote census capability may have applications for firefly conservation.

 This is a nice paper.

 Documenting flash behavior is a challenge. The original use of photomultipliers and analog-to-digital conversion was inefficient. The value of the present study lies in its use of simple video equipment deployed in the field to document flash behavior in real time.

Author reply: Thank you. We hope that our present study can make some contribution to the global firefly studies. 

Major comments:

First, the title is a bit too long. I recommend something like, ¨Species specific flash patterns distinguish sympatric Taiwanese fireflies¨ or "Interflash interval identifies sympatric species in a Taiwanese firefly community.

Author reply: We have changed the title to “Species-Specific Flash Patterns Track the Nocturnal Behavior of Sympatric Taiwanese fireflies”

The paper is too long by 4-5 pages. Much of the results are repetitions of the methods.

 Author reply: In the present manuscript, we usually give a very brief introduction (1 to 2 sentences) on how we design and perform the experiments before describing results, so that the readers can understand our results more easily and clearly. These briefs do not have much overlap with materials and methods.

There appears to be two Figure 2s. I suppose the first one is actually Figure 1.

 Author reply: the figure caption has been corrected. Please check Page 8, Line 270

I recommend a figure to show the time-stack method like a graphic abstract. It still seems to be a frame-by-frame analysis when it is really a stacked image composed of many separate patterns that become superimposed.

  Author reply: We added a supplementary figure for introducing the time-stack method with the description in Methods and Materials. Please check Page 4 Line 156, and the Figure S1 in supplementary files

These species seem to be train flashers, ie they produce long sequences of uniform pulses with equal inter-pulse intervals. If so, how do you know when a pulse train begins or ends? It appears to be assumed that the trains are uniform in timing and intensity from beginning to end.

Author reply: We still face technical difficulties in obtaining a complete luminous signal from a single individual. For instance, according to our observations, a male A. cerata begun to flash when perching, and then started to patrolling fly with a uniform pulse pattern for a few seconds to about 20 minutes. Finally, it landed in the bushes and immediately pause flashing or flashing at a low frequency. The most difficult part to collect a complete signal with handheld camera tracking is that we don't know when and which perching individuals would start flying and landing. It takes luck to get a complete signal. Therefore, we could only collect signals while they were flying, and assume that the pulse train is uniform in time and intensity from beginning to end. Using fixed-point shooting for a long time may be a solution, but the target individual often flies away from the screen, resulting in incomplete signal collection. We are still trying to overcome this problem.

L378: if there are 2 groups, what did your aerial nets reveal? Were samples collected?

 Author reply:  Based on the detailed analysis at 21 April, 2018, we preliminarily deduce that there were two groups active in different nightly periods during the mating season based on the time-dependent change of flash numbers (see the section 3.3, Line 419-432 in the newly PDF version). The subsequent experiments (e.g. species identification, specimen collection) have further shown that these two groups belongs to different species and their nightly active periods were different but partially overlapped (section 3.5, Line 518-528).

Moreover, we also found the similar phenomena during 5 May, 2018 with specimens collection. Both the specimen collection in 21 April and 5 May showed that A. cereta was the dominant species before 21:30, and L. kagiana is the dominant species after 21:30. However, because the specimen information on May 5 is not complete, we did not show it in our present manuscript.

The terminology is imprecise and needs to be standardized. It should follow Lewis and Crastley 2008: flash interval, pulse duration, interpulse interval as the main features to describe flash patterns.

Author reply: We have to apologize for using the imprecise terminology in describing flash pattern due to some reasons. For instance, L. kagiana reveals a clear triple-pulse pattern in the time-stacking image, but it is difficult to differential these three pulses in the Timing diagram, especially the pulse-duration. We could only consider the adjacent triple-pulses of L. kagiana to be a flash signal in the FI and FD measurement. In this embarrassing situation, using ‘pulse-duration’ may confuse the reader that if the pulse-duration denotes one of triple pulses? Thus, we think it would be better to use flash duration (FD) instead of pulse duration in this present study. Moreover, we have added a described ’due to the signal resolution limitation,’ in the Page 9 Line 310-311. Hoping that it will make the description more clearly. Also, due to the signal resolution limitation, we didn’t measure the interpulse interval of the triple-pulse signals.

I have some trouble with the measurements. For instance, FI is measured from peak intensity rather than from pulse onsets. This is problematic. Onsets are when the female response is triggered. Interestingly, on L399 we see the correct use of the term pulse for the three peaks of intensity this species produces.

I do not see how the authors decided where a flash pulse begins or ends. There is no stated threshold for pulse onset or offset. Thus, interflash intervals, pulse durations and pulse intervals can be quite variable.

 Author reply: We initially chose the inter-peak of flashes for flash interval (FI) measurement, which was considering that the peak to peak measurement of flash signals might be much stable than the onset, and it is not easily affected by background noise. According to our newly measurement (roughly analyzed from random selected two individuals from different species), no significant difference was found between inter-peaks and inter-onsets during flash interval measurements in our cases. The p-value between inter-peaks of flashes and inter-onset of flashes is about 0.822 for male A. cerata (n =16, N=2), is about 0.966 (n =20, N=2) for L. kagiana, is about 0.81 for male L. curtithorax (n =19, N=2). We haven’t found that utilizing inter-onset or inter-peak for the measurement will affect our results.

We have defined the measurement of flash duration (FD), which is also the definition of a flash signal (a single-pulse signal or a triple-pulse signal) begin and end (Please see the Page 9 Line 305-309 or iii in Figure 2A-2C). The threshold of a signal onset and offset is same as the FD measurement, about 10% the height of a single-pulse signal or of a triple-pulse signal. Our present study has shown that about 85% of the individuals in a population can be identified by FI pattern matching, suggesting that the FI is still specific enough to identify the species.

It is clear from Figure 2 that cerata and curtithorax are train flashes. Perhaps A. cerata might be considered a single flash species at about 1 hertz. The other two species have flash patterns at 2 and around 7 hertz. However, the intensity diagrams show that L. kagiana actually has a triple-pulsed flash, delivering 3 pulses at 2 hertz! This is too fast for the human eye to detect but the camera captures the peak emissions of each pulse.So, can one detect the flight path from these stacked images?

  Author reply: Yes, we can roughly recognize the triple-pulsed flight path in the time-stacking, please see the Figure 2B and Figure 4A-clock time 22:16. To clear discern the triple-pulse pattern, the frame rate while camera shooting must set at 60 fps or higher.

However, as we mentioned in the discussion (please see Page 24 Line 603-605), it is difficult to discern the triple-pulsed while the individual in slowing flight or too far from the camera or in different flying directions

L133-134: This is not clear. Was recoding continuous or just for a few minutes?

  Author reply: we have corrected the sentence:” To collect the population flash activity, the flashes lasting about 2 min were recorded every 15–60 min from sunset until midnight with fixed-point camera shooting. ” Please check Page 3 Line 136-138.

L215: glad to see this step taken!

 Author reply: Appreciate that. We believe that when conducting field research, it is the duty of every field researcher to reduce the impact on the habitat. We hope to see those fireflies every year instead of letting them disappear after our study.

L238: what is an arbitrary unit? shouldn't this be relative intensity?

 Author reply: According to mention of FIJI/ImageJ introdution, the intensity is measured based on gray-level per pixel of a region of interest (ROI) in a graph, which has a level from 0 to 255. However, it is not correspondence to any physical unit. That is reason why we used arbitrary unit. We didn’t used any comparative light-source or reference, and thus, it is not relative intensity.

L255: shouldn't this be patrolling flight path or signal flight path? This phase of courtship is generally referred to as "patrolling" when male broadcast their identity as versus "courtship" once a female responds and it located. It appears to be sigmoid.

 Author reply: Our current research doesn’t have enough solid evidence to confirm whether it is a patrol flight path or just a simple translocation. We hope that if we can illustrate this point through more video-imagery analysis in future studies. It is difficult to know if the flight path is sigmoid-like while using handheld camera shooting. However, shooting with a fixed-point camera may illustrate this point. In Figure 1A, those flight paths look like sigmoid patterns. This will be a good direction for us to further understand more about the patrolling behaviors in the future.

L353:where is the data for females?

  Author reply: We are not sure about the female data you mentioned. Except the flash pattern of flying females of A. cerata, all female data have been shown in the Figure 3 and Table 2. During our study period, we did not succeed in collecting the female flight signals. In addition, we only collected very few female signals of L. kagiana and L. curtithorax (only one or none per species). We hope to collect enough information about female patterns and present them to everyone in the future.

Table 1: the third columns should be labeled Time(s) (X±SD)

  Author reply: The title of the columns has been changed to ‘Ave Time’, please check Table 1 and Table 2

Flashes produced by perching males likely have no signal value. They may be trophic flashes, ie those produced while coordinating the complex physiology of the flash-control mechanism. This might be mentioned as reason for the lack of significant differences among species.

  Author reply: Totally agree that, there is no stable pattern for those perching signals. One of our guesses is that the perching signals might also be a warning signal to predators.

I have to question the appropriateness and reliability of the time stacking function. If the objective is to measure the flash pattern, why stack flashes? Why not simply measure the onset/offset of each pulse and the time between pulses?

 Author reply: We need the time-stacking image to initial visual flash pattern discrimination, locate the light spot distribution, and further construct the quantitative flash patterns (Timing diagram) for further measurement of FI and FD. For single individual flash studies, the time-stacking method is very important and useful for us to initially identify flash pattern (e.g number of pulses per signal) of individuals in the image before using the timing diagram to establish the flash pattern (or pulse sequence), and it can also be compared with the pattern of the timing diagram for consistency. For population flash studies, the time-stacking method is of great significance in initially identifying and locating the flight path or flash clusters of each flying or perching individual in a population, so that we can easily establish the flash pattern of each individual. We believe that visually image recognition (time stacking) and quantitative flash pattern (timing diagram) are equally important, and it is worth showing both results.

Further, intensity varies according to the inverse square law whereby intensity falls by the square of distance from the source. Moreover, intensity varies according to the direction of the abdomen during flight. Therefore it is wholly unreliable as a species-specific marker of any kind. However, peak intensity indicates pulse number.

  Author reply: Totally agree with you. There is also the reason why the accuracy (85%) of flash interval is much higher than that (41%) of the flash duration in the species identification.

I think the references could better reflect the state of knowledge. For instance, the authors make no mention of the role of sexual selection or predation in shaping flash communication. Variability in flash behavior is the chief means by which females choose mates. This variation bears on the stereotypy of the flash dialogue and thus on the usefulness of flash behavior as a means of species-specific diagnosis. Moreover, courtdships have different phases such as patrol, male-male interactions, female dialoging, and scramble competition-all involving different flash patterns (see Vencl and Carlson 1998).

 Author reply: Very appreciate for sharing the key information with us. Understanding the variability of flashing behavior is a very difficult process, unless we know every details of their flash languages. It might take years to accumulate enough flashing information to be able to distinguish flashing patterns involving courtship, patrols, male and female interactions, female dialogues, and competitions at different phases. Our current research is just the beginning of understanding the possible courtship behavior of these Taiwanese fireflies. We truly hope that we can share more details regarding to their courtship behaviors, after we collect and analyze more flash information from field in the future.

Minor comments:

Abstract:

L25: change determined to "measured" or perhaps "quantified"

 Author reply: Already corrected, please check Page 1 Line 27 “Both FI and FD were quantified from the continuous ….“

L28: change identifying to "distinguishing" or even "differentiating"

 Author reply: Already corrected, please check Page 1 Line 30“ …making them suitable as a reference for differentiating firefly species “

Introduction:

L67: Actually, a great deal more is known about asynchronous species, mainly because there are many, many more asynchronous species, many of which are very common. Correspondingly these have attracted much more attention and we know more about them.

 Author reply: Totally agree with you. The courtship behavior of many asynchronous species has been well studied. However, we found very little information about the group behavior of these asynchronous species, especially the Asian fireflies. We hope that our current research can make some efforts to reveal the group behavior of asynchronous species. Let the world know more about them.

Round 2

Reviewer 3 Report

Dear authors. Thank you for responding to my comments.